# Large Language Diffusion Models

## Abstract

Autoregressive models (ARMs) are widely regarded as the cornerstone of large language models (LLMs). We challenge this notion by introducing **LLaDA**, a diffusion model trained from scratch under the pre-training and supervised fine-tuning (SFT) paradigm. LLaDA models distributions through a forward data masking process and a reverse process, parameterized by a vanilla Transformer to predict masked tokens. By optimizing a likelihood bound, it provides a principled generative approach for probabilistic inference. Across extensive benchmarks, LLaDA demonstrates strong *scalability*, outperforming our self-constructed ARM baselines. Remarkably, LLaDA 8B is competitive with strong LLMs like LLaMA3 8B in *in-context learning* and, after SFT, exhibits impressive *instruction-following* abilities in case studies such as multi-turn dialogue. Moreover, LLaDA addresses the reversal curse, surpassing GPT-4o in a reversal poem completion task. Our findings establish diffusion models as a viable and promising alternative to ARMs, challenging the assumption that key LLM capabilities discussed above are inherently tied to ARMs.[1]

## 1. Introduction

> *What is now proved was once only imagined.*
> —William Blake

Large language models (LLMs) (Zhao et al., 2023) fall entirely within the framework of *generative modeling*. Specifically, LLMs aim to capture the true but unknown language distribution $p_{\text{data}}(\cdot)$ by optimizing a model distribution $p_\theta(\cdot)$ through maximum likelihood estimation, or equivalently KL

[1]Anonymous Institution, Anonymous City, Anonymous Region, Anonymous Country. Correspondence to: Anonymous Author <anon.email@domain.com>.

Preliminary work. Under review by the International Conference on Machine Learning (ICML). Do not distribute.

[1]We will release checkpoints of LLaDA 8B, the inference code and evaluation data after the blind review for reproducibility.

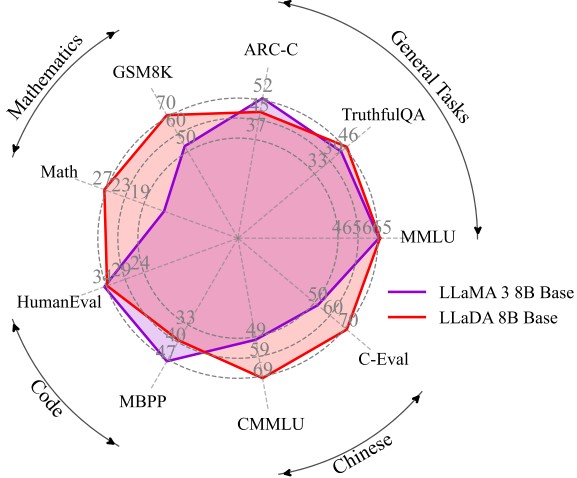

*Figure 1.* **Zero/Few-Shot Benchmarks.** We scale LLaDA to an unprecedented size of 8B parameters from scratch, achieving competitive performance with strong LLMs (Dubey et al., 2024).

divergence minimization between the two distributions:

$$\underbrace{\max_\theta \mathbb{E}_{p_{\text{data}}(x)} \log p_\theta(x) \Leftrightarrow \min_\theta \text{KL}(p_{\text{data}}(x)||p_\theta(x))}_{\text{Generative modeling principles}}. \quad (1)$$

The predominant approach relies on the *autoregressive* modeling (ARM)—commonly referred to as the *next-token prediction* paradigm—to define the model distribution:

$$\underbrace{p_\theta(x) = p_\theta(x^1) \prod_{i=2}^{L} p_\theta(x^i \mid x^1, \dots, x^{i-1})}_{\text{Autoregressive formulation}}, \quad (2)$$

where $x$ is a sequence of length $L$, and $x^i$ is the $i$-th token.

This paradigm has proven remarkably effective (Radford, 2018; Radford et al., 2019; Brown, 2020; OpenAI, 2022) and has become the foundation of current LLMs. Despite its widespread adoption, a fundamental question remains unanswered: *Is the autoregressive paradigm the only viable path to achieving the intelligence exhibited by LLMs?*

We argue that the answer is **not** a simple "yes". The key insight overlooked previously is: it is the **generative modeling principles** (i.e., Eq. (1)), **rather than the autoregressive formulation** (i.e., Eq. (2)) itself, that fundamentally

underpin the essential properties of LLMs, as detailed below. However, certain inherent limitations of LLMs can be directly traced to their autoregressive nature.

In particular, we argue that *scalability* is primarily a consequence of the interplay between Transformers (Vaswani, 2017), model and data size, and *Fisher consistency*[2] (Fisher, 1922) induced by the generative principles in Eq. (1), rather than a unique result of ARM. The success of diffusion transformers (Bao et al., 2023; Peebles & Xie, 2023) on visual data (Brooks et al., 2024) supports this claim.

Furthermore, the capabilities of *instruction-following* and *in-context learning* (Brown, 2020) appear to be intrinsic properties of all proper *conditional* generative models on structurally consistent linguistic tasks, rather than exclusive advantages of ARMs. In addition, while ARMs can be interpreted as a *lossless data compressor* (Deletang et al.; Huang et al., 2024b), any sufficiently expressive probabilistic model can achieve similar capabilities (Shannon, 1948).

Nevertheless, the autoregressive nature of LLMs presents notable challenges. For example, sequential token-by-token generation incurs high computational costs, and the left-to-right modeling limits effectiveness in reversal reasoning tasks (Berglund et al., 2023). These inherent limitations constrain LLMs in handling longer and more complex tasks.

Motivated by these insights, we introduce **LLaDA**, a **L**arge **L**anguage **D**iffusion with m**A**sking, to investigate whether the capabilities exhibited by LLMs can emerge from generative modeling principles beyond ARMs in Eq. (2), thereby addressing the fundamental question posed earlier. In contrast to traditional ARMs, LLaDA leverages a masked diffusion model (MDM) (Austin et al., 2021a; Ou et al., 2024), which incorporates a discrete random masking process and trains a *mask predictor* to approximate its reverse process. This design enables LLaDA to construct a model distribution with bidirectional dependencies and optimize a lower bound of its log-likelihood, offering an unexplored and principled alternative to existing LLMs.

We adopt the standard pipeline of data preparation, pre-training, supervised fine-tuning (SFT), and evaluation, scaling LLaDA to an unprecedented language diffusion of size 8B. In particular, **LLaDA 8B** was pre-trained from scratch on *2.3 trillion tokens* using *0.13 million H800 GPU hours*, followed by SFT on *4.5 million pairs*. Across diverse tasks, including language understanding, math, code, and Chinese, LLaDA demonstrates the following contributions:

**Scalability.** LLaDA scales effectively up to a computational budget of $10^{23}$ FLOPs, achieving comparable results to self-constructed ARM baselines trained on the same data across

---

[2]It suggests the ability to recover the true data distribution with infinite data, a sufficiently large network and optimal training.

six tasks, e.g., MMLU and GSM8K.

**In-Context Learning.** Remarkably, LLaDA 8B surpasses LLaMA2 7B (Touvron et al., 2023) on nearly all 15 standard zero/few-shot learning tasks while performing on par with LLaMA3 8B (Dubey et al., 2024).

**Instruction-Following.** LLaDA significantly enhances the ability to follow instructions after SFT, as demonstrated in case studies such as multi-turn dialogue.

**Reversal Reasoning.** LLaDA effectively breaks the *reversal curse* (Berglund et al., 2023) with consistent performance across forward and reversal tasks. Notably, it outperforms GPT-4o in a reversal poem completion task.

## 2. Approach

In this section, we introduce the probabilistic formulation[3], along with the pre-training, supervised fine-tuning, and inference procedures for LLaDA, as illustrated in Fig. 2.

### 2.1. Probabistic Formulation

Unlike ARMs in Eq. (2), LLaDA defines a model distribution $p_\theta(x_0)$ through a *forward process* and a *reverse process* (Austin et al., 2021a; Ou et al., 2024). The forward process gradually masks tokens independently in $x_0$ until the sequence is fully masked at $t = 1$. For $t \in (0, 1)$, the sequence $x_t$ is partially masked, with each being masked with probability $t$ or remaining unmasked with probability $1 - t$. The reverse process recovers the data distribution by iteratively predicting masked tokens as $t$ moves from 1 to 0.

The core of LLaDA is a *mask predictor*, a parametric model $p_\theta(\cdot|x_t)$ that takes $x_t$ as input and predicts all masked tokens (denoted M) simultaneously. It is trained using a cross-entropy loss computed only on the masked tokens:

$$\mathcal{L}(\theta) \triangleq -\mathbb{E}_{t,x_0,x_t}\left[\frac{1}{t}\sum_{i=1}^{L}\mathbf{1}[x_t^i = \mathrm{M}]\log p_\theta(x_0^i|x_t)\right], \quad (3)$$

where $x_0$ is sampled from the training data, $t$ is sampled uniformly from $[0, 1]$, and $x_t$ is sampled from the forward process. The indicator function $\mathbf{1}[\cdot]$ ensures that the loss is computed only for masked tokens.

Once trained, we can simulate a reverse process (see Sec. 2.4 for details) parameterized by the mask predictor and define the model distribution $p_\theta(x_0)$ as the marginal distribution induced at $t = 0$. Notably, the loss function in Eq. (3) has been proven to be an upper bound on the negative log-likelihood of the model distribution (Ou et al., 2024):

$$-\mathbb{E}_{p_{\text{data}}(x_0)}\left[\log p_\theta(x_0)\right] \leq \mathcal{L}(\theta), \quad (4)$$

---

[3]Here, we focus on the approach of LLaDA. A rigorous formulation of MDM is provided in Appendix A for interested readers.

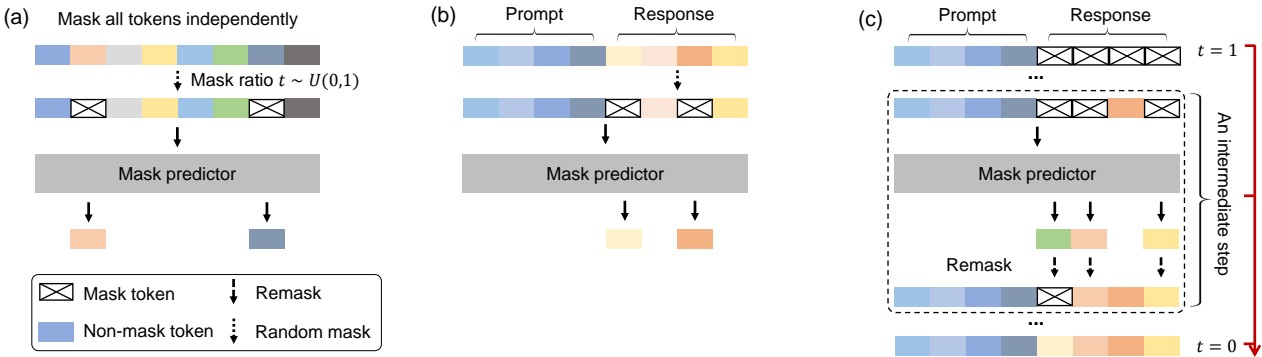

*Figure 2.* **A Conceptual Overview of LLaDA.** (a) Pre-training. LLaDA is trained on text with random masks applied independently to all tokens at the same ratio $t \sim U[0, 1]$. (b) SFT. Only response tokens are possibly masked. (c) Sampling. LLaDA simulates a diffusion process from $t = 1$ (fully masked) to $t = 0$ (unmasked), predicting all masks simultaneously at each step with flexible remask strategies.

making it a principled objective for generative modeling.

Notably, LLaDA employs a masking ratio that varies randomly between 0 and 1 while masked language models (Devlin, 2018) use a fixed ratio. The subtly differences have significant implications, especially at scale: as shown in Eq. (4), LLaDA is a principled generative model with the potential to perform *in-context learning* naturally, akin to LLMs. Moreover, its generative perspective ensures *Fisher consistency* (Fisher, 1922) in extreme cases, suggesting strong *scalability* with large data and models.

## 2.2. Pre-training

LLaDA employs a Transformer (Vaswani, 2017) as the mask predictor, whose architecture is similar to existing LLMs. However, LLaDA does not use a causal mask, as its formulation allows it to see the entire input for predictions.

We trained two variants of LLaDA with different sizes: 1 billion (B) and 8B. We summarize the model architecture of LLaDA 8B and LLaMA3 8B (Dubey et al., 2024) here and details are provided in Appendix B.2. We have ensured consistency in most hyperparameters while making several necessary modifications. We use vanilla multi-head attention instead of grouped query attention (Ainslie et al., 2023) for simplicity, as LLaDA is incompatible with KV caching, resulting in a different number of key and value heads. Consequently, the attention layer has more parameters, and we reduce the FFN dimension to maintain a comparable model size. Additionally, the vocabulary size differs slightly due to a tokenizer (Brown, 2020) adapted on our data.

The LLaDA model is pre-trained on a dataset comprising *2.3 trillion* (T) tokens, adhering to a data protocol that aligns closely with existing large language models (LLMs) (Touvron et al., 2023; Chu et al., 2024), without the incorporation of any special techniques. The data are derived from online corpora, with low-quality content filtered through manually designed rules and LLM-based approaches. Beyond general

text, the dataset encompasses high-quality code, math, and multilingual data. The mixing of data sources and domains is guided by scaled-down ARMs. The pre-training process utilizes a fixed sequence length of 4096 tokens, incurring a total computational cost of *0.13 million H800 GPU hours*, similar to ARMs of the same scale and dataset size.

For a training sequence $x_0$, we randomly sample $t \in [0, 1]$, mask each token independently with the same probability $t$ to obtain $x_t$ (see Fig. 2 (a)) and estimate Eq. (3) via the Monte Carlo method for stochastic gradient descent training. In addition, following Nie et al. (2024), to enhance the ability of LLaDA to handle variable-length data, we set 1% of the pre-training data to a random length that is uniformly sampled from the range $[1, 4096]$.

We adopted the Warmup-Stable-Decay (Hu et al., 2024) learning rate scheduler to monitor the training progress without interrupting continuous training. Specifically, we linearly increased the learning rate from 0 to $4 \times 10^{-4}$ over the first 2000 iterations and maintained it at $4 \times 10^{-4}$. After processing 1.2T tokens, we decayed the learning rate to $1 \times 10^{-4}$ and held it constant for the next 0.8T tokens to ensure stable training. Finally, we linearly reduced the learning rate from $1 \times 10^{-4}$ to $1 \times 10^{-5}$ for the last 0.3T tokens. Furthermore, we utilized the AdamW optimizer (Loshchilov, 2017) with a weight decay of 0.1, a batch size of 1280, and a local batch size of 4 per GPU. The 8B experiment was executed once, without any hyperparameter tuning.

## 2.3. Supervised Fine-Tuning

We enhance the capability of LLaDA to follow instructions by supervised fine-tuning (SFT) with paired data $(p_0, r_0)$, where $p_0$ is the prompt and $r_0$ denotes the response. This is the simplest and most basic post-training method for LLMs. Technically, this requires to model the *conditional distribution* $p_\theta(r_0 | p_0)$ instead of $p_\theta(x_0)$ in pre-training.

The implementation is similar to pre-training. As shown in

Fig. 2 (b), we leave the prompt unchanged and mask the tokens in the response independently, as done for $x_0$. Then, we feed both the prompt and the masked response $r_t$ to the pre-trained mask predictor to compute the loss for SFT:

$$-\mathbb{E}_{t,p_0,r_0,r_t}\left[\frac{1}{t}\sum_{i=1}^{L'}\mathbf{1}[r_t^i = \mathrm{M}]\log p_\theta(r_0^i|p_0, r_t)\right], \quad (5)$$

where $L'$ denotes a dynamic length specified later, and all other notations remain the same as before.

Note that this approach is fully compatible with pre-training. Essentially, the concatenation of $p_0$ and $r_0$ can be treated as clean pre-training data $x_0$, while the concatenation of $p_0$ and $r_t$ serves as the masked version $x_t$. The process is identical to pre-training, with the only difference being that all masked tokens happen to appear in the $r_0$ portion.

The LLaDA 8B model undergoes SFT on a dataset comprising *4.5 million* pairs. Consistent with the pre-training process, both data preparation and training follow the SFT protocols utilized in existing LLMs (Chu et al., 2024; Yang et al., 2024), without introducing any additional techniques to optimize LLaDA's performance. The dataset spans multiple domains, including code, mathematics, instruction-following, and structured data understanding. We append |EOS| tokens to the end of short pairs in each mini-batch to ensure equal lengths across all data. We treat |EOS| as a normal token during training and remove it during sampling, enabling LLaDA to control the response length automatically. Please refer to Appendix B.1 for more details.

We train for 3 epochs on the SFT data using a similar schedule to the pre-training phase. The learning rate is linearly increased from 0 to $2.5 \times 10^{-5}$ over the first 50 iterations and then kept constant. During the final $10\%$ of iterations, it is linearly reduced to $2.5 \times 10^{-6}$. Additionally, we set the weight decay to 0.1, the global batch size to 256, and the local batch size to 2 per GPU. The SFT experiment was executed once, without any hyperparameter tuning.

### 2.4. Inference

As a generative model, LLaDA is capable of both sampling new text and evaluating the likelihood of candidate text.

We begin with the sampling. As illustrated in Fig. 2 (c), given a prompt $p_0$, we discretize the reverse process to sample from the model distribution $p_\theta(r_0|p_0)$, starting from a fully masked response. The total number of sampling steps is a hyperparameter, which naturally provides LLaDA with a trade-off between efficiency and sample quality, as analyzed in Sec. 3.3. We employ uniformly distributed timesteps by default. In addition, the generation length is also treated as a hyperparameter, specifying the length of the fully masked sentence at the beginning of the sampling process. As de-

tailed in Appendix B.4, since both pre-training and SFT are conducted using datasets with variable lengths, the final results are insensitive to this length hyperparameter.

At an intermediate step from time $t \in (0, 1]$ to $s \in [0, t)$, we feed both $p_0$ and $r_t$ into the mask predictor and predict all masked tokens simultaneously. Subsequently, we *remask* $\frac{s}{t}$ of the predicted tokens in expectation to obtain $r_s$, ensuring that the transition of the reverse process aligns with the forward process for accurate sampling (Austin et al., 2021a).

In principle, the remasking strategy should be purely random. However, inspired by the annealing tricks of sampling in LLMs (Holtzman et al., 2019; Brown, 2020), we explore two deterministic yet effective remasking strategies. Specifically, similarly to Chang et al. (2022), we remask the $\frac{s}{t}$ of predicted tokens with the lowest confidence based on the predictions, called *low-confidence* remasking. Additionally, for LLaDA after SFT, we can divide the sequence into several blocks and generate them from left to right, called *semi-autoregressive* remasking. Within each block, we apply the reverse process to perform sampling. We provide more details and ablation studies in Appendix. B.3.

For conditional likelihood evaluation, we can naturally utilize the upper bound in Eq. (5). However, we find that the following equivalent form (Ou et al., 2024) exhibits lower variance and is more stable for evaluation:

$$-\mathbb{E}_{l,r_0,r_l}\left[\frac{L}{l}\sum_{i=1}^{L}\mathbf{1}[r_l^i = \mathrm{M}]\log p_\theta(r_0^i|p_0, r_l)\right], \quad (6)$$

where $l$ is uniformly sampled from $\{1, 2, \ldots, L\}$, and $r_l$ is obtained by uniformly sampling $l$ tokens from $r_0$ without replacement for masking. In addition, we employ the unsupervised classifier-free guidance (Nie et al., 2024). We refer the readers to more details in Appendix A.2.

We present the training, sampling, and likelihood evaluation algorithms, along with theoretical details, in Appendix A.

## 3. Experiments

We evaluate the scalability, instruction-following, and in-context learning capabilities of LLaDA on standard benchmarks, followed by analyses and case studies on more controlled datasets to provide a comprehensive assessment.

### 3.1. Scalability of LLaDA on Language Tasks

We first investigate the *scalability* of LLaDA on downstream tasks in comparison with the ARM baselines we constructed. Specifically, at the 1B scale, we ensured that LLaDA and ARM shared the same architecture, data, and all other configurations. At larger scales, we also report results for LLaDA and ARM models of slightly different sizes trained on the same data due to resource limit, as detailed in Appendix B.2.

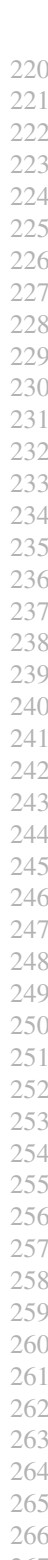
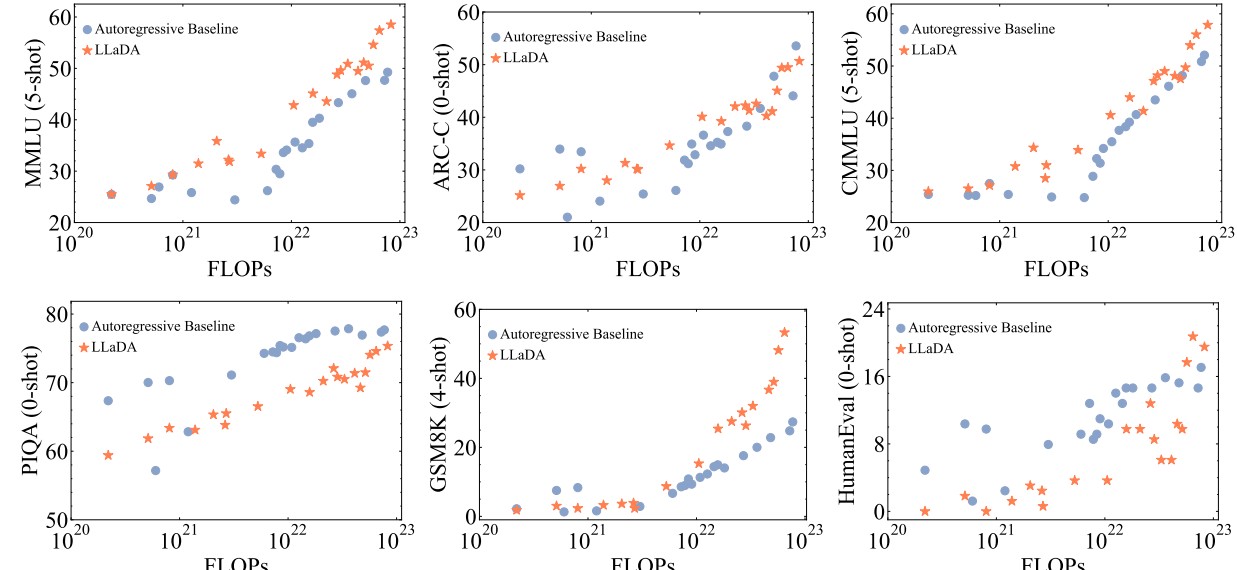

*Figure 3.* **Scalability of LLaDA.** We evaluate the performance of LLaDA and our ARM baselines trained on the same data across increasing computational FLOPs. LLaDA exhibits strong scalability, matching the overall performance of ARMs on six tasks.

We use the computational cost as a unified scaling metric. For evaluation, we focused on six standard and diverse tasks.

As shown in Fig. 3, LLaDA demonstrates impressive scalability, with its overall trend being highly competitive with that of ARM. Notably, in tasks such as MMLU and GSM8K, LLaDA exhibits even stronger scalability. Even on tasks like PIQA, where performance lags, LLaDA narrows the gap with ARMs at larger scales. To account for the significant influence of outliers, we opted not to fit quantitative scaling curves, avoiding potential misinterpretation. Nevertheless, the results clearly demonstrate the scalability of LLaDA.

Nie et al. (2024) suggests that MDM requires 16 times more computation than ARM to achieve the same likelihood. However, there are key differences that make the conclusions of this study more broadly applicable. In particular, likelihood is a relatively indirect metric for downstream task performance, and diffusion optimizes a bound of the likelihood, making it not directly comparable to ARM. Additionally, we extended the scaling range from $10^{18} \sim 10^{20}$ in Nie et al. (2024) to $10^{20} \sim 10^{23}$ in this work.

### 3.2. Benchmark Results

To comprehensively evaluate the *in-context learning* and *instruction-following* capabilities of LLaDA 8B, we conducted detailed comparisons with existing LLMs (Touvron et al., 2023; Dubey et al., 2024; Chu et al., 2024; Yang et al., 2024; Bi et al., 2024; Jiang et al., 2023) of similar scale. The selection of tasks and evaluation protocols was aligned with existing studies, encompassing 15 popular benchmarks in general tasks, mathematics, code, and Chinese. Further

details are provided in Appendix B.5. For a more direct comparison, we re-evaluated representative LLMs (Touvron et al., 2023; Dubey et al., 2024) in our implementation.

As shown in Tab. 1, after pretraining on 2.3T tokens, LLaDA 8B demonstrates remarkable performance, surpassing LLaMA2 7B on nearly all tasks, and is overall competitive with LLaMA3 8B. LLaDA shows advantages in math and Chinese tasks. We conjecture that the strengths stem from the same factors as its relatively weaker performance in some tasks—differences in data quality and distribution, largely due to the closed-source situation of LLM datasets.

Notably, we have carefully ruled out the possibility of data leakage by taking GSM8K as an example. First, as shown in Fig. 3, LLaDA outperformed ARM baselines regarding GSM8K. Moreover, the conclusion remains on a fully unseen GSM8K-like task (Ye et al., 2024) in Appendix B.7.

Further, Tab. 2 compares the performance of LLaDA 8B Instruct with existing LLMs. We observed that SFT improved LLaDA's performance on most downstream tasks. A few metrics, such as MMLU, showed declines, and we conjecture may be due to the suboptimal quality of the SFT data. Overall, since we did not perform alignment with reinforcement learning (RL), our results are slightly behind LLaMA3 8B Instruct, though the gaps in many metrics remain small. Notably, even with only SFT, LLaDA demonstrates impressive instruction-following abilities, as detailed in Sec. 3.4. We leave RL-based alignment for future work.

Overall, despite the lack of data transparency, we have made every effort to adopt standardized procedures and introduce

*Table 1.* **Benchmark Results of Pre-trained LLMs.** * indicates that LLaDA 8B Base, LLaMA2 7B Base, and LLaMA3 8B Base are evaluated under the same protocol, detailed in Appendix B.5. Results indicated by † and ¶ are sourced from Chu et al. (2024); Yang et al. (2024) and Bi et al. (2024) respectively. The numbers in parentheses represent the number of shots used for evaluation. "-" indicates unknown data.

| | LLaDA 8B* | LLaMA3 8B* | LLaMA2 7B* | Qwen2 7B† | Qwen2.5 7B† | Mistral 7B† | Deepseek 7B¶ |
|---|---|---|---|---|---|---|---|
| Model | Diffusion | AR | AR | AR | AR | AR | AR |
| Training tokens | 2.3T | 15T | 2T | 7T | 18T | - | 2T |
| *General Tasks* | | | | | | | |
| MMLU | **65.9** (5) | 65.4 (5) | 45.9 (5) | 70.3 (5) | 74.2 (5) | 64.2 (5) | 48.2 (5) |
| BBH | 49.8 (3) | **57.6** (3) | 37.3 (3) | 62.3 (3) | 70.4 (3) | 56.1 (3) | 39.5 (3) |
| ARC-C | 47.9 (0) | **53.1** (0) | 46.3 (0) | 60.6 (25) | 63.7 (25) | 60.0 (25) | 48.1 (0) |
| Hellaswag | 72.5 (0) | **79.1** (0) | 76.0 (0) | 80.7 (10) | 80.2 (10) | 83.3 (10) | 75.4 (0) |
| TruthfulQA | **46.4** (0) | 44.0 (0) | 39.0 (0) | 54.2 (0) | 56.4 (0) | 42.2 (0) | - |
| WinoGrande | 74.8 (5) | **77.3** (5) | 72.5 (5) | 77.0 (5) | 75.9 (5) | 78.4 (5) | 70.5 (0) |
| PIQA | 74.4 (0) | **80.6** (0) | 79.1 (0) | - | - | - | 79.2 (0) |
| *Mathematics & Science* | | | | | | | |
| GSM8K | **70.7** (4) | 53.1 (4) | 14.3 (4) | 80.2 (4) | 85.4 (4) | 36.2 (4) | 17.4 (8) |
| Math | **27.3** (4) | 15.1 (4) | 3.2 (4) | 43.5 (4) | 49.8 (4) | 10.2 (4) | 6.0 (4) |
| GPQA | **26.1** (5) | 25.9 (5) | 25.7 (5) | 30.8 (5) | 36.4 (5) | 24.7 (5) | - |
| *Code* | | | | | | | |
| HumanEval | 33.5 (0) | **34.2** (0) | 12.8 (0) | 51.2 (0) | 57.9 (0) | 29.3 (0) | 26.2 (0) |
| HumanEval-FIM | **73.8** (2) | 73.3 (2) | 26.9 (2) | - | - | - | - |
| MBPP | 38.2 (4) | **47.4** (4) | 18.4 (4) | 64.2 (0) | 74.9 (0) | 51.1 (0) | 39.0 (3) |
| *Chinese* | | | | | | | |
| CMMLU | **69.9** (5) | 50.7 (5) | 32.5 (5) | 83.9 (5) | - | - | 47.2 (5) |
| C-Eval | **70.5** (5) | 51.7 (5) | 34.0 (5) | 83.2 (5) | - | - | 45.0 (5) |

*Table 2.* **Benchmark Results of Post-trained LLMs.** LLaDA only employs an SFT procedure while other models have extra reinforcement learning (RL) alignment. * indicates that LLaDA 8B Instruct, LLaMA2 7B Instruct, and LLaMA3 8B Instruct are evaluated under the same protocol, detailed in Appendix B.5. Results indicated by † and ¶ are sourced from Yang et al. (2024) and Bi et al. (2024) respectively. The numbers in parentheses represent the number of shots used for in-context learning. "-" indicates unknown data.

| | LLaDA 8B* | LLaMA3 8B* | LLaMA2 7B* | Qwen2 7B† | Qwen2.5 7B† | Gemma2 9B† | Deepseek 7B¶ |
|---|---|---|---|---|---|---|---|
| Model | Diffusion | AR | AR | AR | AR | AR | AR |
| Training tokens | 2.3T | 15T | 2T | 7T | 18T | 8T | 2T |
| Post-training | SFT | SFT+RL | SFT+RL | SFT+RL | SFT+RL | SFT+RL | SFT+RL |
| Alignment pairs | 4.5M | - | - | 0.5M + - | 1M + 0.15M | - | 1.5M + - |
| *General Tasks* | | | | | | | |
| MMLU | 65.5 (5) | **68.4** (5) | 44.1 (5) | - | - | - | 49.4 (0) |
| MMLU-pro | 37.0 (0) | **41.9** (0) | 4.6 (0) | 44.1 (5) | 56.3 (5) | 52.1 (5) | - |
| Hellaswag | 74.6 (0) | **75.5** (0) | 51.5 (0) | - | - | - | 68.5 (-) |
| ARC-C | **88.5** (0) | 82.4 (0) | 57.3 (0) | - | - | - | 49.4 (-) |
| *Mathematics & Science* | | | | | | | |
| GSM8K | **78.6** (4) | 78.3 (4) | 29.0 (4) | 85.7 (0) | 91.6 (0) | 76.7 (0) | 63.0 (0) |
| Math | 26.6 (0) | **29.6** (0) | 3.8 (0) | 52.9 (0) | 75.5 (0) | 44.3 (0) | 15.8 (0) |
| GPQA | 31.8 (5) | **31.9** (5) | 28.4 (5) | 34.3 (0) | 36.4 (0) | 32.8 (0) | - |
| *Code* | | | | | | | |
| HumanEval | 47.6 (0) | **59.8** (0) | 16.5 (0) | 79.9 (0) | 84.8 (0) | 68.9 (0) | 48.2 (-) |
| MBPP | 34.2 (4) | **57.6** (4) | 20.6 (4) | 67.2 (0) | 79.2 (0) | 74.9 (0) | 35.2 (-) |

*Table 3.* **Comparison in the Poem Completion Task**.

|  | Forward | Reversal |
| --- | --- | --- |
| GPT-4o (2024-08-06) | **82.7** | 34.3 |
| Qwen2.5 7B Instruct | 75.9 | 38.0 |
| LLaDA 8B Instruct | 48.8 | **42.4** |

diverse tasks, we believe they sufficiently demonstrate the extraordinary capabilities of LLaDA, which is the only competitive non-autoregressive model to our knowledge.

### 3.3. Reversal Reasoning and Analyses

To quantify the reversal reasoning ability of models (Berglund et al., 2023), we follow the protocol established in Allen-Zhu & Li (2023). Specifically, we construct a dataset of 496 famous Chinese poem sentence pairs. Given a sentence from a poem, models are tasked with generating the subsequent line (forward) or the preceding line (reversal) without additional fine-tuning. Examples can be found in Appendix B.8. This setting provides a straightforward and more realistic evaluation compared to previous studies (Nie et al., 2024; Kitouni et al., 2024).

As shown in Tab. 3, LLaDA effectively addresses the *reversal curse* (Berglund et al., 2023), demonstrating consistent zero-shot performance across both forward and reversal tasks. In contrast, both Qwen 2.5 and GPT-4o exhibit a significant gap between the two. The results on forward generation confirm that both ARMs are strong, benefiting from significantly larger datasets and greater computational resources than LLaDA. However, LLaDA outperforms both by a large margin in the reversal task.

We emphasize that we did not design anything special for reversal tasks. Intuitively, LLaDA treats tokens uniformly without inductive bias, leading to balanced performance. See more details in Appendix A.2.

We also analyze the effect of remasking strategies and sampling steps, as detailed in Appendix B.3 and Appendix B.6.

### 3.4. Case Studies

We present samples generated by LLaDA 8B Instruct in Tab. 4, showcasing its instruction-following capabilities. First, the table illustrates LLaDA's ability to generate coherent, fluent, and extended text in a non-autoregressive manner. Second, it highlights the model's multi-turn dialogue capability, effectively retaining conversation history and producing contextually appropriate responses across multiple languages. Such *chat* capabilities of LLaDA are impressive, as it departs from conventional ARMs for the first time, to the best of our knowledge. See more case studies on remasking and reasoning tasks in Appendix B.9.

## 4. Related Work

Diffusion models (Sohl-Dickstein et al., 2015; Ho et al., 2020; Song et al., 2020) have excelled in visual domains but remain unverified for LLMs despite extensive efforts.

A simple approach is to continuousize text data and apply diffusion models directly (Li et al., 2022; Gong et al., 2022; Han et al., 2022; Strudel et al., 2022; Chen et al., 2022; Dieleman et al., 2022; Richemond et al., 2022; Wu et al., 2023; Mahabadi et al., 2024). Alternatively, some methods model continuous parameters of discrete distributions instead (Lou & Ermon, 2023; Graves et al., 2023; Lin et al., 2023; Xue et al., 2024). However, scalability remains a challenge, as a 1B parameter model requires *64 times the compute* of an ARM to achieve comparable performance (Gulrajani & Hashimoto, 2024).

Another approach replaces continuous diffusion with discrete processes featuring new forward and reverse dynamics (Austin et al., 2021a), leading to numerous variants (Hoogeboom et al., 2021b;a; He et al., 2022; Campbell et al., 2022; Meng et al., 2022; Reid et al., 2022; Sun et al., 2022; Zheng et al., 2023; Chen et al., 2023; Gat et al., 2024; Zheng et al., 2024; Sahoo et al., 2024; Shi et al., 2024). Notably, Lou et al. (2023) showed that masked diffusion, as a special case of discrete diffusion, achieves perplexity comparable to or surpassing ARMs at GPT-2 scale. Ou et al. (2024) established fundamental theoretical results, which motivated our model design, training, and inference (see Appendix A). Nie et al. (2024) explored how MDM can be leveraged for language tasks such as question answering at GPT-2 scale. Gong et al. (2024) fine-tune ARMs in the MDM formulation. However, improvements are confined to certain metrics, and it remains unclear whether this approach can yield a foundation model comparable to strong LLMs under a comprehensive evaluation.

In comparison, this study scales MDM to an unprecedented size of 8B parameters from scratch, achieving performance comparable to leading LLMs such as LLaMA 3.

Notably, a parallel line of work on image generation (Chang et al., 2022; 2023) aligns with MDMs applied to text data.

## 5. Conclusion and Discussion

> *In the middle of difficulty lies opportunity.*
> —Albert Einstein

We introduce **LLaDA**, a principled and previously unexplored approach to large language modeling based on diffusion models. LLaDA demonstrates strong capabilities in scalability, in-context learning, and instruction-following, achieving performance comparable to strong LLMs. In addition, LLaDA offers unique advantages such as bidirectional

*Table 4.* **Visualization of the Sampling Process and a Generated Multi-round Dialogue.** In the response of LLaDA, darker colors indicate tokens predicted in the later stages of sampling, while lighter colors correspond to earlier predictions.

---

*Sampling Process*

| User | Lily can run 12 kilometers per hour for 4 hours. After that, she runs 6 kilometers per hour. How many kilometers can she run in 8 hours? |
|------|------|

LLaDA

In  4  hours  ,  Lily  runs  12  *  4  =  48  kilometers  .  After  4

hours  she  runs  6  *  4  =  24  kilometers  .  In  total  she  runs  48

+  24  =  72  kilometers  in  8  hours  .  Result  72

---

*Multi-round Dialogue Case*

| User | Could you please share with me what the initial two lines of the renowned poem 'The Road Not Taken'? |
|------|------|
| LLaDA | Certainly! The initial two lines of the renowned poem "The Road Not Taken" by Robert Frost are: "Two roads diverged in a yellow wood, And sorry I could not travel both." These lines set the stage for the poem, introducing the idea of a speaker facing a choice between two paths. |
| User | Please help me translate into Chinese. |
| LLaDA | Sure! The first two lines of "The Road Not Taken" by Robert Frost can be translated into Chinese as: "两条路分岔在黄色的树林中，遗憾我不能同时走" |
| User | Please translate into German. |
| LLaDA | Sure! The first two lines of "The Road Not Taken" by Robert Frost can be translated into German as: "Zwei Wege trennten sich im gelben Wald, und leider konnte ich nicht beide tun." |
| User | Please also write a poem about life choices. I require 5 sentences, and the first word of each sentence starts with C as the first letter. |
| LLaDA | Certainly! Here's a poem about life choices, with each sentence starting with the letter "C": Choices cascade through life's journey, Creating connections with each decision, Careful consideration leads to better paths, Courageous choices shape our destiny, Choosing wisely guides us through the years. |

---

modeling and enhanced robustness, effectively addressing several inherent limitations of existing LLMs. Our findings not only *establish diffusion models as a viable and promising alternative* but also *challenge the prevailing assumption that these essential capabilities are inherently tied to ARMs*.

While promising, the full potential of diffusion models remains to be fully explored. Several limitations of this work present significant opportunities for future research.

Due to computational constraints, direct comparisons between LLaDA and ARMs—such as training on identical datasets—were restricted to a computational budget of less than $10^{23}$ FLOPs. To allocate resources for training the largest possible LLaDA model and showcasing its potential, we were unable to scale the ARM baseline to the same extent. Moreover, no specialized attention mechanisms or position embeddings were designed for LLaDA, nor were any system-level architectural optimizations applied. On the inference side, our exploration of guidance mecha-

nisms (Dhariwal & Nichol, 2021; Ho & Salimans, 2022) remains preliminary, and LLaDA currently exhibits sensitivity to inference hyperparameters. Furthermore, LLaDA has yet to undergo alignment with reinforcement learning (Ouyang et al., 2022; Rafailov et al., 2024), which is crucial for improving its performance and alignment with human intent.

Looking forward, the scale of LLaDA is still smaller than leading counterparts (Achiam et al., 2023; Dubey et al., 2024; Google, 2024; Anthropic, 2024; Yang et al., 2024; Liu et al., 2024), highlighting the need for further scaling to fully assess its capabilities. In addition, LLaDA's ability to handle multi-modal data remains unexplored. The impact of LLaDA on prompt tuning techniques (Wei et al., 2022) and its integration into agent-based systems (Park et al., 2023; Wang et al., 2024) is yet to be fully understood. Finally, a systematic investigation into post-training for LLaDA could facilitate the development of O1-like systems (OpenAI, 2024; Guo et al., 2025).

## Impact Statement

This work challenges the prevailing dominance of autoregressive modeling in LLMs by proposing a diffusion-based alternative. Our findings open new avenues for exploring alternative probabilistic paradigms in natural language processing, with potential applications in conversational AI, code generation, and complex reasoning tasks.

However, diffusion-based models, like traditional LLMs, raise similar societal concerns. These include the environmental impact of large-scale training, the potential misuse for generating harmful or misleading content, and the amplification of biases present in training data. Addressing these challenges is critical to ensuring the responsible development and deployment of diffusion-based language models.

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

---

**Algorithm 1** Pre-training of LLaDA

---

**Require:** mask predictor $p_\theta$, data distribution $p_{\text{data}}$

1: **repeat**
2:    $x_0 \sim p_{\text{data}}, t \sim \text{U}(0,1)$           # with a probability of 1%, the sequence length of $x_0$ follows U[1, 4096]
3:    $x_t \sim q_{t|0}(x_t|x_0)$           # $q_{t|0}$ is defined in Eq. (7)
4:    Calculate $\mathcal{L} = -\frac{1}{t*L} \sum_{i=1}^{L} \mathbf{1}[x_t^i = \text{M}] \log p_\theta(x_0^i|x_t)$       # $L$ is the sequence length of $x_0$
5:    Calculate $\nabla_\theta \mathcal{L}$ and run optimizer.
6: **until** Converged
7: **Return** $p_\theta$

---

**Algorithm 2** Supervised Fine-Tuning of LLaDA

---

**Require:** mask predictor $p_\theta$, pair data distribution $p_{\text{data}}$

1: **repeat**
2:    $p_0, r_0 \sim p_{\text{data}}, t \sim \text{U}(0,1)$        # please refer to Appendix B.1 for details on the SFT data processing.
3:    $r_t \sim q_{t|0}(r_t|r_0)$          # $q_{t|0}$ is defined in Eq. (7)
4:    Calculate $\mathcal{L} = -\frac{1}{t*L'} \sum_{i=1}^{L'} \mathbf{1}[r_t^i = \text{M}] \log p_\theta(r_0^i|p_0, r_t)$      # $L'$ is the sequence length of $r_0$
5:    Calculate $\nabla_\theta \mathcal{L}$ and run optimizer.
6: **until** Converged
7: **Return** $p_\theta$

---

**Algorithm 3** Conditional Log-likelihood Evaluation of LLaDA

---

**Require:** mask predictor $p_\theta$, prompt $p_0$, response $r_0$, the number of Monte Carlo estimations $n_{mc}$

1: log_likelihood = 0
2: **for** $i \leftarrow 1$ to $n_{mc}$ **do**
3:    $l \sim \{1, 2, \ldots, L\}$          # $L$ is the sequence length of $r_0$
4:    Obtain $r_l$ by uniformly sampling $l$ tokens from $r_0$ without replacement for masking
5:    log_likelihood = log_likelihood $+ \frac{L}{l} \sum_{i=1}^{L} \mathbf{1}[r_l^i = \text{M}] \log p_\theta(r_0^i|p_0, r_l)$
6: **end for**
7: log_likelihood = log_likelihood$/n_{mc}$
8: **Return** log_likelihood

---

# A. Formulation of Masked Diffusion Models

## A.1. Training

Masked diffusion models (MDMs) (Austin et al., 2021a; Lou et al., 2023; Ou et al., 2024) define the model distribution $p_\theta(x_0)$ in a manner distinct from autoregressive models.

These models introduce a forward process $\{x_t\}$ indexed by a time $t \in [0, 1]$. This process gradually and independently masks all tokens in the sequence $x_0$. At time $t = 0$, the data point $x_0$ is fully observed with no masks, while for $t \in (0, 1]$, $x_t$ represents latent variables with varying mask ratios in expectation.

Formally, the conditional distribution of $x_t$ given $x_0$ is defined by a fully factorized form:

$$q_{t|0}(x_t|x_0) = \prod_{i=1}^{L} q_{t|0}(x_t^i|x_0^i), \tag{7}$$

where the conditional distribution for each token is given by:

$$q_{t|0}(x_t^i|x_0^i) = \begin{cases} 1 - t, & x_t^i = x_0^i, \\ t, & x_t^i = \text{M}. \end{cases} \tag{8}$$

Here, M denotes the mask token. Intuitively, each token either remains unchanged or is masked, with the probability of

---

**Algorithm 4** Reverse Process of LLaDA

---

**Require:** mask predictor $p_\theta$, prompt $p_0$, answer length $L$, sampling steps $N$
  1: Set $r_1$ is a fully masked sequence of length $L$.
  2: **for** $t \leftarrow 1$ **down to** $\frac{1}{N}$ **step** $\frac{1}{N}$ **do**
  3:    $s = t - \frac{1}{N}$
  4:    $r_0 = \arg\max_{r_0} p_\theta(r_0|p_0, r_t)$                    # we employ greedy sampling when predicting masked tokens
  5:    **for** $i \leftarrow 1$ **to** $L$ **do**
  6:       **if** $r_t \neq \text{M}$ **then**
  7:          $r_0^i = r_t^i$
  8:       **else**
  9:          With probability $\frac{s}{t}$, $r_0^i$ is set to M
 10:       **end if**
 11:    **end for**
 12:    $r_s = r_0$
 13: **end for**
 14: **Return** $r_0$

---

being masked increasing linearly as $t$ progresses from 0 to 1. At $t = 1$, all tokens are guaranteed to be masked, meaning that $x_1$ follows a Dirac distribution concentrated on a sequence of fully masked tokens. Notably, the linear masking probability is analogous to but distinct from, the noise schedule in continuous diffusion models (Sohl-Dickstein et al., 2015; Ho et al., 2020; Song et al., 2020). This linearity is motivated by the assumption that the information in the text is proportional to the number of tokens on average, making it reasonable to lose information linearly during the forward process.

The forward process is not only reversible but also corresponds to a reverse process that is fully factorized across all tokens (Austin et al., 2021a). The reverse process, from time $t = 1$ to 0, generates new data from sequences of fully masked tokens. The conditional distribution for the reverse process, for $0 \leq s < t \leq 1$, is factorized as:

$$q_{s|t}(x_s|x_t) = \prod_{i=1}^{L} q_{s|t}(x_s^i|x_t), \tag{9}$$

where the conditional distribution for each token is:

$$q_{s|t}(x_s^i|x_t) = \begin{cases} 1, & x_t^i \neq \text{M}, \ x_s^i = x_t^i, \\ \frac{s}{t}, & x_t^i = \text{M}, \ x_s^i = \text{M}, \\ \frac{t-s}{t} q_{0|t}(x_s^i|x_t), & x_t^i = \text{M}, \ x_s^i \neq \text{M}, \\ 0, & \text{otherwise.} \end{cases} \tag{10}$$

Thus, the key function to estimate is the conditional distribution $q_{0|t}(x_s^i|x_t)$, which predicts the original token if it is masked in the input $x_t$. This is analogous to the *data prediction* form in continuous diffusion models.

As proven in (Ou et al., 2024), an equivalent yet *time-free* parameterization can be derived as:

$$q_{0|t}(x_s^i|x_t) = p_{\text{data}}(x_0^i|x_t^{\text{UM}}), \quad \forall i \text{ such that } x_t^i = \text{M}, \tag{11}$$

where $x_t^{\text{UM}}$ denotes the collection of unmasked tokens in $x_t$, which is identical to the corresponding tokens in the original data $x_0$ since unmasked tokens are solely determined by $x_0$ and are independent of time $t$. Intuitively, this implies that estimating the data prediction function is equivalent to estimating the conditional distributions on clean data, which is time-invariant. Consequently, the time $t$ need not be provided as input to the parametric model.

Although the development of masked diffusion is nontrivial, the implementation is straightforward. We first introduce the *mask predictor*, a parametric model $p_\theta(\cdot|x_t)$ (e.g., a Transformer without a causal mask), which takes $x_t$ for any $t$ as input and predict all masked tokens simultaneously. Then, we define the model distribution $p_\theta(x_0)$ as follows: starting with $x_1$ as a sequence of fully masked tokens, we simulate an approximate reverse process parameterized by $p_\theta(\cdot|x_t)$ from $t = 1$ to 0. The marginal distribution induced at $t = 0$ then represents the model distribution $p_\theta(x_0)$.

---

**Algorithm 5** Low-confidence Remasking Strategy of LLaDA

---

**Require:** mask predictor $p_\theta$, prompt $p_0$, answer length $L$, sampling steps $N$

1: Set $r_1$ is a fully masked sequence of length $L$.
2: **for** $t \leftarrow 1$ **down to** $\frac{1}{N}$ **step** $\frac{1}{N}$ **do**
3:     $s = t - \frac{1}{N}$
4:     **for** $i \leftarrow 1$ to $L$ **do**
5:       **if** $r_t^i \neq \mathrm{M}$ **then**
6:         $r_0^i = r_t^i, c^i = 1$
7:       **else**
8:         $r_0^i = \arg\max_{r_0^i} p_\theta(r_0^i | p_0, r_t)$
9:         $c^i = p_\theta(r_0^i | p_0, r_t)_{r_0^i}$
10:      **end if**
11:     **end for**
12:     $n_{un} = \lfloor L(1-s) \rfloor$                                 # the number of unmasked tokens is $n_{un}$ in timestep $s$
13:     **for** $i \leftarrow 1$ to $L$ **do**
14:       **if** $c^i \in \text{Lowest} - n_{un}\left(\{c^i\}_1^L\right)$ **then**
15:         $r_0^i = \mathrm{M}$                 # the $n_{un}$ positions with the least confidence are selected for remasking.
16:       **end if**
17:     **end for**
18:     $r_s = r_0$
19: **end for**
20: **Return** $r_0$

---

Formally, the mask predictor is trained using a cross-entropy loss with masking:

$$\mathcal{L}(\theta) \triangleq -\mathbb{E}_{t,x_0,x_t}\left[\frac{1}{t}\sum_{i=1}^{L}\mathbf{1}[x_t^i = \mathrm{M}]\log p_\theta(x_0^i | x_t)\right], \tag{12}$$

where $x_0$ is sampled from the training data, $t$ is sampled uniformly from $[0, 1]$, and $x_t$ is sampled from $q_{t|0}(x_t | x_0)$. The indicator function $\mathbf{1}[\cdot]$ ensures that the cross-entropy loss is computed only for masked tokens. In Ou et al. (2024), it has been proven that the loss function $\mathcal{L}(\theta)$ is an upper bound on the negative log-likelihood of the model distribution:

$$-\mathbb{E}_{x_0 \sim p_{\text{data}(x_0)}}\left[\log p_\theta(x_0)\right] \leq \mathcal{L}(\theta). \tag{13}$$

In summary, this principled approach trains a generative model by progressively masking tokens during a forward process and learning to recover the data distribution during a reverse process, all under the (approximate) maximum likelihood estimation framework.

### A.2. Inference

The cross-entropy loss in Eq. (12) has several equivalent forms (Ou et al., 2024). The first one is given by

$$-\mathbb{E}_{l \sim \{1,2,\ldots,L\},x_0,x_l}\left[\frac{L}{l}\sum_{i=1}^{L}\mathbf{1}[x_l^i = \mathrm{M}]\log p_\theta(x_0^i | x_l)\right], \tag{14}$$

where $l$ is uniformly sampled from $\{1, 2, \ldots, L\}$, and $x_l$ is obtained by uniformly sampling $l$ tokens from $x_0$ without replacement for masking. Despite masking exactly $l$ tokens is different from masking each token independently with probability $t$, these two masking methods lead to equivalent results in expectation (Ou et al., 2024).

While Eq. (12) and Eq. (14) share the same expectation, their variances differ. Intuitively, in Eq. (12), we expect $x_t$ to have a fraction of $t$ tokens masked. However, the randomness of the forward process (i.e., Eq. (7)) often causes deviations, especially when $x_t$ contains few tokens. In contrast, in Eq. (14), the fraction of masked tokens in $x_l$ is deterministically $\frac{l}{L}$. While a theoretical analysis depends on the data distribution, empirical results show that Eq. (12) requires over 1000 Monte Carlo estimates for stable results, whereas Eq. (14) achieves stability with only 128 estimates. In addition, we can simply modify Eq. (14) to its conditional version (i.e., Eq. (6)) based on Eq. (5).

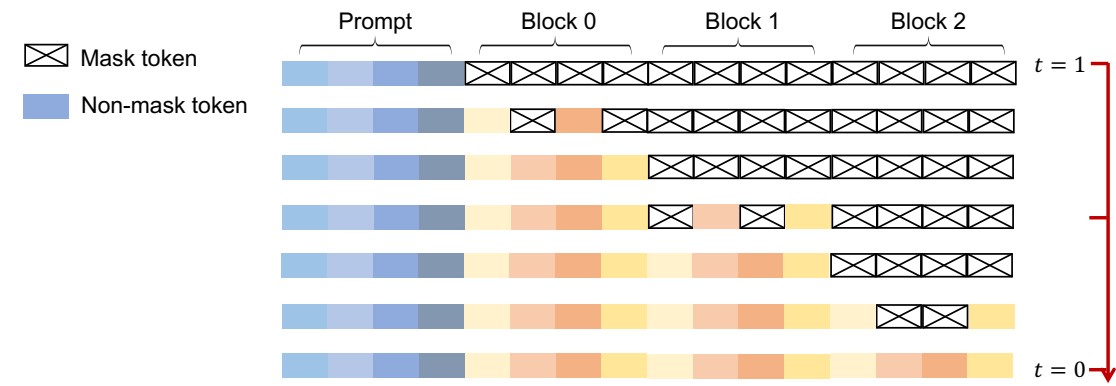

*Figure 4.* **A Conceptual Overview of the Semi-autoregressive Sampling.**

Any-order autoregressive models (AO-ARM) (Uria et al., 2014; Hoogeboom et al., 2021a; Shih et al., 2022) characterize the joint distribution autoregressively for all possible orders $\pi$ of the $L$ variables. To learn such a distribution, an AO-ARM utilizes a weight-sharing neural network to model all univariate conditionals and employs mask tokens to represent absent variables. During training, the expected negative log-likelihood over the uniform distribution of all orders $U_\pi$ is minimized:

$$-\mathbb{E}_{x_0, \pi \sim U_\pi} \left[ \sum_{i=1}^{L} \log p_\theta(x_0^{\pi(i)} | x_0^{\pi(<i)}; \pi) \right]. \tag{15}$$

Intuitively, $x_0^{\pi(<i)}$ can be understood as a masked token $x_t$ with index in $\pi(\geq i)$ being masked. It can be further proved that Eq. (15) is equivalent to Eq. (12). This connection explains the bidirectional reasoning capabilities of LLaDA, even though it was never used explicitly in the inference procedure.

Nie et al. (2024) introduce unsupervised classifier-free guidance, a plug-and-play technique that balances alignment with prompts and text diversity. Specifically, unsupervised classifier-free guidance employs the following modified mask predictor for inference:

$$\tilde{p}_\theta(r_0 | p_0, r_t) \propto \frac{p_\theta(r_0 | p_0, r_t)^{1+w}}{p_\theta(r_0 | m, r_t)^w}, \tag{16}$$

where $m$ is a mask sequence of the same length as $p_0$ and $w$ is a hyperparameter that controls the strength of $p_0$. We employ unsupervised classifier-free guidance in the downstream tasks, as detailed in Appendix B.5.

### A.3. Algorithms

In this section, we present the training and inference algorithms. Specifically, we introduce the pre-training and supervised fine-tuning algorithms in Algorithm 1 and Algorithm 2, respectively. In addition, the likelihood evaluation algorithm is provided in Algorithm 3. Finally, we present the sampling algorithms in Algorithm 4 and Algorithm 5, which correspond to the original reverse process (i.e., random remasking) and the low-confidence remasking strategy (Chang et al., 2022) of LLaDA, respectively.

Furthermore, we apply a semi-autoregressive remasking strategy for the instruct model. Specifically, the sequence is divided into several blocks and generated from left to right. Within each block, we apply the random remasking or the low-confidence remasking strategy for sampling. For clarity and better visualization, we illustrate the semi-autoregressive sampling process in Fig. 4.

Our semi-autoregressive remasking strategy is inspired by the prior work of Anonymous (2025). However, it is important to note that Anonymous (2025) and LLaDA have entirely different training objectives and network architectures. Specifically, the pre-training objective of Anonymous (2025) differs from Eq. (3). Moreover, their approach incorporates a causal mask in the Transformer attention block, whereas LLaDA employs a bidirectional Transformer.

*Table 5.* **Model Architecture.** We provide the configurations of our 1B and 7B ARM baselines, as well as the 1B and 8B LLaDA and the 8B LLaMA3 models.

| | Our ARM Baseline 1B | LLaDA 1B | Our ARM Baseline 7B | LLaDA 8B | LLaMA3 8B |
|---|---|---|---|---|---|
| Layers | 22 | 22 | 28 | 32 | 32 |
| Model dimension | 2048 | 2048 | 4096 | 4096 | 4096 |
| Attention heads | 32 | 32 | 32 | 32 | 32 |
| Vocabulary size | 126,464 | 126,464 | 126,464 | 126,464 | 128,000 |
| FFN dimension | 5634 | 5634 | 13,440 | 12,288 | 14,336 |
| Key/Value heads | 4 | 4 | 8 | 32 | 8 |
| Total parameters | 1.49 B | 1.49 B | 6.83 B | 8.02 B | 8.03 B |
| Non-embedding parameters | 0.97 B | 0.97 B | 5.80 B | 6.98 B | 6.98 B |

## B. Experiments

### B.1. Details of SFT Data

A dynamic sequence length is employed during supervised SFT, where |EOS| tokens are appended to the end of short pairs to ensure equal sequence lengths across all data in each mini-batch. Notably, the padding |EOS| tokens are treated as part of the response, i.e., masked and included in the training objective. The |EOS| tokens are removed from the generated outputs during sampling. This strategy is crucial for LLaDA and ensures that the model learns to control the length of its responses by generating |EOS|, enabling the response length to align effectively with the given prompt.

In addition, for $n$-turn dialogues $(p_0^0, r_0^0, p_0^1, r_0^1, \ldots, p_0^{n-1}, r_0^{n-1})$, we treat it as $n$ single-turn dialogue pairs, i.e., $(p_0^0, r_0^0), (p_0^0 r_0^0 p_0^1, r_0^1), \ldots, (p_0^0 r_0^0 p_0^1 r_0^1 \ldots p_0^{n-1}, r_0^{n-1})$ and randomly sample one. This data partitioning strategy not only equips LLaDA with multi-turn dialogue capabilities but also aligns with the padding strategy.

### B.2. Architectures and Scaling Experiments

This section provides detailed information about the models we trained and the scalability experiments described in Sec. 3.1.

Firstly, for efficiency, we trained an ARM and a MDM, both with 1.5B parameters and identical architectures. Additionally, we scaled the MDM to 8B parameters. Due to computational resource constraints, we did not train an 8B autoregressive model with the same architecture. Instead, we utilized our previously trained 7B autoregressive model for comparison. These four models are utilized in the scalability analysis in Sec. 3.1.

We adopted a Transformer architecture similar to LLaMA (Touvron et al., 2023; Dubey et al., 2024) for the ARMs and MDMs we trained. Specifically, we employ RMSNorm (Zhang & Sennrich, 2019) to stabilize training, used SwiGLU (Shazeer, 2020) as the activation function to enhance non-linearity, and integrate RoPE (Su et al., 2024) for more expressive positional encoding. Tab. 5 provides an overview of the model architectures.

Additionally, we employ the widely used $6ND$ formulation (Kaplan et al., 2020; Hoffmann et al., 2022) to calculate the training FLOPs in Fig. 3, where $N$ represents the number of non-embedding parameters, and $D$ denotes the total number of training tokens.

### B.3. Details and Ablation on Remasking

This section first explains the inference methods in detail, followed by ablation studies on various remasking strategies.

For instruct model, since our training includes a large number of |EOS| tokens, directly applying the lowest confidence remasking strategy would result in an excessively high and unnatural occurrence of |EOS| tokens in the generated sentences. Therefore, we divide the entire generation length into several blocks. Each block is generated from left to right, and within each block, we apply the reverse process to perform sampling. However, for the base model, since |EOS| tokens are not padded during training, using the semi-autoregressive remasking strategy makes little difference in the results.

*Table 6.* **Ablation on Remasking.** We use GSM8K accuracy as the evaluation metric. The lowest-confidence & semi-autoregressive remasking strategy refers to generating each block from left to right, while applying the lowest-confidence remasking strategy within each block.

|  | LLaDA 8B Base | LLaDA 8B Instruct |
|---|---|---|
| Randomly remasking | 52.3 | 72.0 |
| Lowest confidence remasking | **64.7** | 12.9 |
| Lowest confidence & semi-autoregressive remasking | 64.4 | **73.8** |

*Table 7.* **Ablation on Generated Length.** We use the GSM8K accuracy as the evaluation metric and set the number of sampling steps to half of the generated length.

| Length | LLaDA 8B Base | LLaDA 8B Instruct |
|---|---|---|
| 256 | 62.5 | 75.3 |
| 512 | 64.7 | 73.8 |
| 1024 | 65.9 | 75.3 |

For efficiency, we set the answer length to 512 and the number of sampling steps to 256 in our ablation studies. For the semi-autoregressive remasking strategy, we set the block length to 32. To simplify the evaluation, we use GSM8K accuracy as the performance metric. As shown in Table 6, for the base model, applying the lowest confidence remasking strategy significantly improves performance, whereas the semi-autoregressive remasking strategy does not yield any noticeable benefits. In contrast, for the instruct model, directly applying the lowest confidence remasking strategy leads to a substantial performance drop. However, combining the lowest confidence remasking strategy with the semi-autoregressive remasking strategy results in the best performance for the instruct model.

We provide the specific remasking strategies and hyperparameters used for each benchmark in Appendix B.5.

### B.4. Ablation on Generated Length

In this section, we conduct ablation studies on the generated length. For simplicity, we use GSM8K as the evaluation metric. To ensure efficiency and fairness, for each setting, we set the number of sampling steps to half of the generated length, ensuring that in each sampling step, two tokens are transferred from the mask to the text.

For the LLaDA 8B Base model, we employ the lowest-confidence remasking strategy (i.e., Algorithm 5). For the LLaDA 8B Instruct model, we adopt semi-autoregressive sampling with a block length of 32.

As reported in Tab. 7, both the Base model and the Instruct model exhibit minimal sensitivity to the length hyperparameter.

### B.5. Standard Benchmarks and Evaluation

In this section, we briefly introduce the benchmarks used in our study and describe the details of our evaluation process.

Following standard LLM (Chu et al., 2024; Yang et al., 2024) evaluation practices, we assess LLaDA across four dimensions:

**General ability:** MMLU (Hendrycks et al., 2020), BBH (Suzgun et al., 2022), ARC-C (Clark et al., 2018), Hellaswag (Zellers et al., 2019), TruthfulQA (Lin et al., 2021), WinoGrande (Sakaguchi et al., 2021) and PIQA (Bisk et al., 2020).

**Math and science ability:** GSM8K (Cobbe et al., 2021), Math (Hendrycks et al., 2021) and GPQA (Rein et al., 2023).

**Code generation:** HumanEval (Chen et al., 2021), HumanEval-FIM (Bavarian et al., 2022) and MBPP (Austin et al., 2021b).

**Chinese understanding:** CMMLU (Li et al., 2023) and C-Eval (Huang et al., 2024a).

For all the aforementioned benchmarks, we follow the widely adopted evaluation process (Gao et al., 2024) used in LLM assessments, primarily employing conditional likelihood estimation and conditional generation. Specifically, for certain benchmarks, a prompt and multiple candidate answers are provided, and the model is required to compute each candidate's

*Table 8.* **Inference Configuration for LLaDA 8B Instruct.** We apply the low-confidence remasking strategy across all datasets. A block length smaller than the answer length indicates the use of the semi-autoregressive remasking strategy; otherwise, it is not applied. The shorter answer lengths for MMLU and HellaSwag result from their requirement to generate only a single token (i.e., A, B, C, or D) and two special end tokens (i.e., end of the model response and end of the sentence).

|          | Answer length | Block length |
| -------- | ------------- | ------------ |
| MMLU     | 3             | 3            |
| Hellaswag | 3            | 3            |
| GPQA     | 128           | 64           |
| GSM8K    | 256           | 8            |
| HumanEval | 512          | 32           |
| MBPP     | 512           | 32           |
| Math     | 256           | 256          |
| MMLU-pro | 256           | 256          |
| ARC-C    | 512           | 512          |

*Table 9.* **Comparison on iGSM Dataset.**

|               | 4 steps | 5 steps | 6 steps |
| ------------- | ------- | ------- | ------- |
| LLaMA3 8B Base | 38.0   | 35.0    | 34.0    |
| LLaDA 8B Base  | **64.0** | **41.0** | **44.0** |

conditional likelihood. The candidate with the highest likelihood is then selected as the model's final answer, and accuracy is used as the evaluation metric. For the remaining benchmarks, the model generates responses based on the given prompt, and performance is evaluated using metrics such as exact match and other relevant criteria.

For the base model, we use conditional likelihood estimation for MMLU, CMMLU, C-Eval, ARC-C, Hellaswag, TruthfulQA, WinoGrande, PIQA, and GPQA, while the remaining benchmarks are evaluated using conditional generation. For the instruct model, we evaluate all benchmarks using conditional generation. For datasets evaluated with conditional likelihood estimation, we adopt the widely used lm-evaluation-harness (Gao et al., 2024) framework. However, for datasets evaluated with conditional generation, we rely on an internal library for evaluation, as lm-evaluation-harness lacks implementations for certain key metrics (e.g., HumanEval-FIM).

For benchmarks evaluated using conditional likelihood estimation, we use Monte Carlo estimation to approximate Eq. (6) for conditional likelihood estimation. Since MMLU, CMMLU, and C-EVAL only require the likelihood of a single token, a single Monte Carlo estimate is sufficient for these benchmarks. For all other benchmarks, we find that 128 Monte Carlo samples are adequate to produce stable results. For ARC-C, Hellaswag, TruthfulQA, WinoGrande, PIQA, and GPQA, we report the results with the optimal unsupervised classifier-free guidance scale searched in $\{0, 0.5, 1, 1.5, 2\}$.

For benchmarks evaluated using conditional generation, we apply the lowest confidence remasking strategy to the base model, setting both the response length and the number of sampling steps to 1024. For the instruct model, we set the sampling steps equal to the answer length and provide the inference hyperparameters in Table 8. Due to computational resource constraints, we do not use the unsupervised classifier-free guidance for these benchmarks evaluated using conditional generation.

### B.6. Analysis of Sampling Steps

In this section, we analyze the impact of sampling steps. For simplicity, we use the LLaDA 8B Base model and evaluate it on the GSM8K and HumanEval datasets. Additionally, we fix the answer length at 1024. As shown in Fig. 5, as the number of sampling steps increases to match the answer length, the model's accuracy improves accordingly.

### B.7. Evaluation on iGSM Dataset

To further assess the mathematical capabilities of LLaDA, we test its performance on iGSM (Ye et al., 2024), an infinite, synthetic GSM8K-like dataset. iGSM is generated via specific rules, with parameters that control the difficulty of problems (i.e., the number of solution steps). For evaluation consistency, we append "#### $answer" to the final solution, adhering to

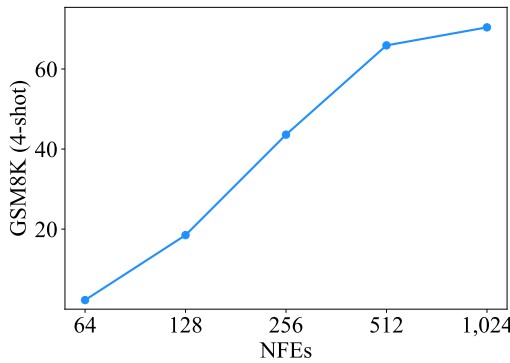 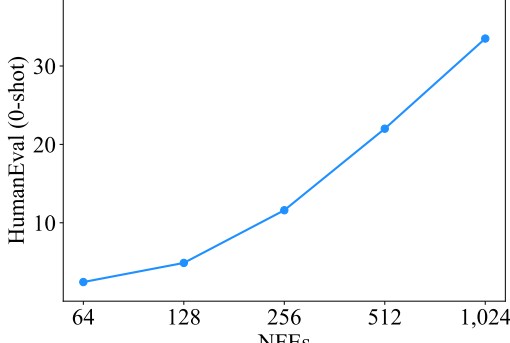

*Figure 5.* **Analysis of Sampling Steps.**

the GSM8K format. Below is an example with solution steps set to 4:

---

(**Question**) The number of each North Star Elementary's Cultural Studies Classroom equals 1. The number of each Westridge Elementary's Dance Studio equals 3 times as much as the sum of each North Star Elementary's Classroom and each North Star Elementary's Cultural Studies Classroom. How many Dance Studio does Westridge Elementary have?

(**Solution**) Define North Star Elementary's Cultural Studies Classroom as x; so x = 1.
Define North Star Elementary's Classroom as m; so m = x = 1.
Define Westridge Elementary's Dance Studio as n; w = m + x = 1 + 1 = 2;
so n = 3 * w = 3 * 2 = 1 #### 1

---

Since there are slight differences between GSM8K and iGSM (e.g., the use of a mod 5 algorithmic system), we follow (Ye et al., 2024) and provide a system prompt along with four-shot question-answer pairs for each problem.

---

(**Prompt**) You're an expert at solving elementary math problems involving addition, subtraction, and multiplication. You solve all the problems in a uniform format. All calculations are done modulo 5. For example, 4 + 4 equals 3, 2 + 4 equals 1, 3 + 3 + 3 equals 4, 3 * 3 equals 4, and 2 * 2 equals 4. When providing your solution, please end with '#### x.' where x is your final answer, an integer between 0 and 4. You must solve all the problems using the same solution format. Our scenarios involve up to four categories of objects: schools, classrooms, backpacks and stationeries. Each school may contain classrooms, each classroom may contain backpacks, and each backpack may contain stationeries. We can specify quantities, such as ẗhe number of dance studios at each Lakeshore High.Ässume that every entity with the same name has an identical configuration; for example, each Lakeshore High contains the same number of dance studios. Another guiding principle is that what is not mentioned does not exist: when we refer to classrooms at Lakeshore High, we are only discussing the classrooms explicitly mentioned in our scenario. Furthermore, if Lakeshore High is not even mentioned, any classroom within it is automatically considered to be non-existent (i.e. 0).

---

For solution steps ranging from 4 to 6, we generate 100 questions for each case and report the corresponding accuracy in Table 9. As shown in the table, LLaDA Base demonstrates significant and consistent advantages over LLaMA3 on unseen mathematical problems, aligning with the results in Table 1.

### B.8. Poem Completion Tasks

In this section, we present examples from our poem completion dataset as follows.

Example 1:
Prompt: 窈窕淑女的下一句是什么？直接输出句子即可。
Answer: 君子好逑。

*Table 10.* **Visualization of the Semi-Autoregressive Sampling Process.** In the response of LLaDA, darker colors indicate tokens predicted in the later stages of sampling, while lighter colors correspond to earlier predictions.

| User | What are the benefits of regular exercise for physical and mental health? |
|------|--------------------------------------------------------------------------|
| LLaDA | Regular exercise can improve physical health by strengthening muscles and bones, improving cardiovascular health, and reducing the risk of chronic diseases such as diabetes and heart disease. It can also improve mental health by reducing stress and anxiety, improving mood, and enhancing cognitive function. Additionally, exercise can improve sleep quality and increase energy levels. |

*Table 11.* **Visualization of the Multi-turn Dialogue.** We employ random remasking strategy. In the response of LLaDA, darker colors indicate tokens predicted in the later stages of sampling, while lighter colors correspond to earlier predictions.

| User | I want to have a pet, but I don't know whether to get a cat or a dog. Do you have any suggestions? |
|------|----------------------------------------------------------------------------------------------------|
| LLaDA | It really depends on your preferences, lifestyle, and the type of pet you want. If you want an independent pet and have a busy schedule, a cat is a good choice. If you prefer a companion and have a more active lifestyle, a dog might be a better fit. Have you considered your options? |
| User | I'm very busy at work, so it seems like a cat would be a more suitable option. Do you have any suggestions for getting a cat? |
| LLaDA | Sure! First, make sure you have enough space in your home and provide food, water, and toys for your cat. Also, research different breeds and personalities, and choose cats that are suitable for your lifestyle. Lastly, make sure you have the necessary vaccinations and insurance in place before you get a pet. |

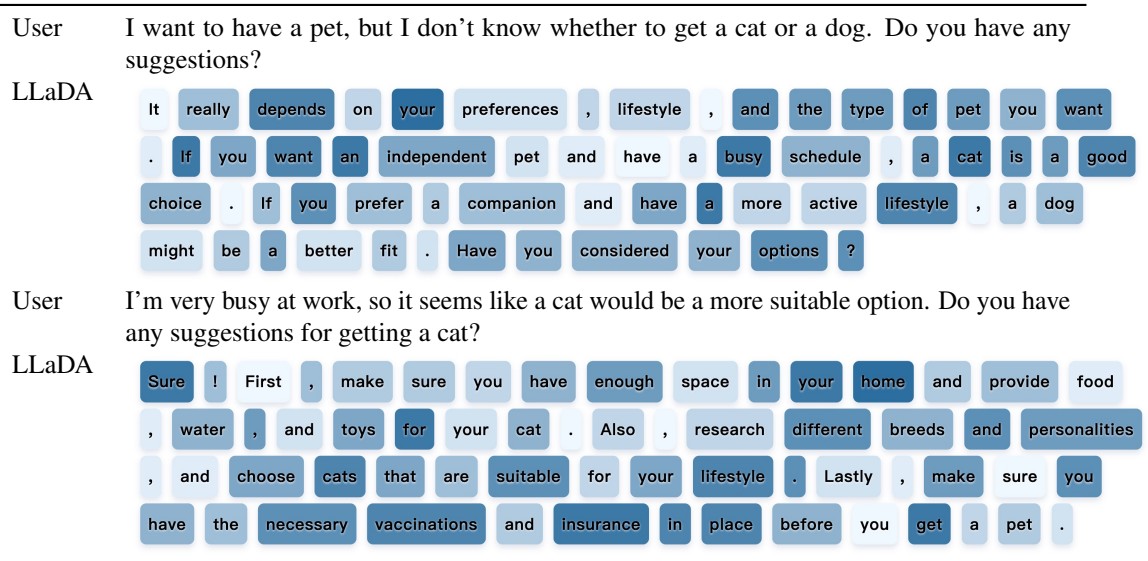

Example 2:

Prompt: 不拘一格降人才的上一句是什么？直接输出句子即可。

Answer: 我劝天公重抖擞。

## B.9. More Case Studies

In this section, we present additional case studies of LLaDA 8B Instruct. First, Tab. 10 illustrates the sampling process of semi-autoregressive remasking, while Tab. 11 depicts the sampling process for multi-turn dialogues with random remasking. Additionally, Tab. 12 and Tab. 13 provide further examples of single-turn and multi-turn dialogues. Finally, Tab. 14 presents examples of poem reversal completions where the LLaDA 8B Instruct model succeeds, in contrast to the failure of GPT-4o.

*Table 12.* **Single-turn Dialogue Cases of LLaDA 8B Instruct.**

| | | |
|---|---|---|
| *Daily life* | | |
| | User | What's a piece of advice you've received that you've found helpful? |
| | LLaDA | One helpful piece of advice I've received is to always be open to learning and growth. I've found that being willing to explore new ideas and experiences has helped me become more resilient and adaptable. By embracing a growth mindset, I've been able to identify opportunities for improvement and development, both personally and professionally. |
| *Daily life* | | |
| | User | Please recommend me three famous movies |
| | LLaDA | Certainly! Here are three famous movies that have achieved widespread popularity and critical acclaim: |
| | | 1. **The Shawshank Redemption** (1994) - Inspired by a novella and directed by Frank Darabont, this film tells the story of Andy Dufresne, a man convicted of murdering his wife, dealing with his harsh life in Shawshank Prison and forming an unlikely friendship with a fellow inmate. |
| | | 2. **The Godfather** (1972) - Directed by Francis Ford Coppola, this film is based on Mario Puzo's novel of the same name. It follows the story of the Corleone family, particularly Michael Corleone, who rises to power and takes over the family's criminal empire. |
| | | 3. **The Empire Strikes Back** (1980) - Directed by George Lucas, this film continues the story of a young Skywalker in the Star Wars saga. It follows Luke Skywalker as he helps the Rebel Alliance fight against the Galactic Empire and destroy the powerful weapon, the Death Star. |
| | | All of these films are celebrated for their cinematic achievements, powerful storytelling, and memorable characters. |
| *Translation* | | |
| | User | Please help me translate into Chinese: 'What is now proved was once only imagined' |
| | LLaDA | 现在被证明的曾经只是想象 |
| *Translation* | | |
| | User | Please help me translate into German: 'What is now proved was once only imagined' |
| | LLaDA | Wer heute bestätigt ist, war einst nur immaginiert. |
| *Math* | | |
| | User | Emma decides to run 4 sprints 5 times a week. She runs 75 meters each sprint. How many total meters does she run a week? |
| | LLaDA | Emma runs 4 sprints each time, so she runs a total of 4 x 5 = 20 sprints a week. Each sprint is 75 meters long, so Emma runs a total of 20 x 75 = 1500 meters a week. The final result is 1500 |
| *Code* | | |
| | User | I am writing python code, I have an array, please help me sort it from largest to smallest |
| | LLaDA | Certainly! You can sort an array from the largest to smallest in Python using the 'sort()' method or the 'sorted()' function. Here's an example: |
| | | ### Using 'sort()' method: |
| | | my_array = [3, 1, 4, 1, 5, 2] |
| | | my_array.sort(reverse=True) |
| | | print(my_array) |

*Table 13.* **Multi-turn Dialogue Cases of LLaDA 8B Instruct.**

| *Daily life* | |
| --- | --- |
| User | How do I prepare for a job interview? |
| LLaDA | To prepare for a job interview, you should research the company and the job, practice common interview questions, and dress appropriately. |
| User | What are some common interview questions? |
| LLaDA | Some common interview questions include "Tell me about yourself" and "Why do you want to work for us"? |

| *Daily life* | |
| --- | --- |
| User | How do I start learning programming? |
| LLaDA | To learn programming, you can start by choosing a programming language that you are interested in. Some popular languages include Python, Java, and C++. |
| User | Which programming language should I learn first? |
| LLaDA | It depends on your interests and goals. Python is a great language for beginners for its simplicity and versatility. |
| User | How long does it take to learn Python? |
| LLaDA | It can take anywhere from a few weeks to several months to learn Python. |
| User | What are some good Python projects for beginners? |
| LLaDA | Some good Python projects for beginners include building a simple calculator, a game, and a web application. |

| *Math* | |
| --- | --- |
| User | Natalia sold clips to 48 of her friends in April, and then she sold half as many clips in May. How many clips did Natalia sell altogether in April and May? |
| LLaDA | Natalia sold clips to 48 friends in April. She sold half as many clips in May, so she sold $48/2 = 24$ clips in May. Altogether, Natalia sold $48 + 24 = 72$ clips in April and May. The final result is 72. |
| User | Natalia sold three times as many items in June as she did in May. How many clips did Natalia sell altogether in these three months? |
| LLaDA | Natalia sold clips to 48 friends in April. She sold half as many clips in May, so she sold $48/2 = 24$ clips in May. Natalia sold three times as many items in June as she did in May, so she sold $24 * 3 = 72$ clips in June. Altogether, Natalia sold $48 + 24 + 72 = 144$ clips in these three months. The final result is 144. |

*Table 14.* **Poem Reversal Completion Cases where LLaDA 8B Instruct Succeeds but GPT-4o Fails.**

| *Question1* | |
| --- | --- |
| User | '但闻人语响'的上一句是什么? |
| GPT-4o | "但闻人语响"的上一句是"无可奈何花落去",这两句出自唐代词人李清照的（武陵春,春晚: 无可奈何花落去，但闻人语响。这首词表达了作者在春天时节的孤独与感伤，借花落和人语声的描写，反映出词人内心的无亲与失落。 |
| LLaDA | "但闻人语响"的上一句是"空山不见人" |

| *Question2* | |
| --- | --- |
| User | '夜静春山空'的上一句是什么? |
| GPT-4o | "夜静春山空"的上一句是"空山新雨后".这两句出自唐代诗人王维的（山居秋膜）:空山新雨后，夜静春山空。这首诗描绘了春天山中雨后的寂静与空灵，表现出自然的宁静与和谐美。 |
| LLaDA | "夜静春山空"的上一句是"人闲桂花落" |

