# OpenReview forum: "Large Language Diffusion Models"
_ICML.cc/2025/Conference — Submitted to ICML 2025_

### Official Review · Reviewer_8p3t · 2025-03-11

**Overall Recommendation:** 4

**Summary:**

This work presents the large language diffusion model (LLaDA), a masked diffusion that shows strong scalability outperforming self-constructed autoregressive large language models. In particular, LLaDA achieves comparable performance for in-context learning and instruction-following compared to SOTA LLMs, and outperforms on a reversal reasoning task. Furthermore, the paper showcases the chat capability of LLaDA supporting multi-round dialogue which was known only possible for autoregressive models. While few papers studied the scalability of diffusion models, LLaDA is the first to show the promising capability of diffusion language model comparable with autoregressive language models on multiple benchmarks while supporting multi-round dialogue chat ability.

**Claims And Evidence:**

Yes, the claims on the scalability and other capabilities like in-context learning, and instruction following are supported by experimental results.

**Essential References Not Discussed:**

To the best of my knowledge, most of the relevant works on diffusion language models and their scalability are addressed in this work.

**Experimental Designs Or Analyses:**

Yes, the experimental designs and analyses are valid and appropriate.

**Methods And Evaluation Criteria:**

Yes, the proposed methods and evaluation criteria including experiment design, baseline selection, and benchmark datasets and tasks are all appropriate and well-designed.

**Other Comments Or Suggestions:**

No additional comments.

**Other Strengths And Weaknesses:**

**Strength**

- Comprehensively demonstrated the scalability of diffusion models and their capabilities in in-context learning, instruction-following, and reversal reasoning on benchmark datasets.

- First to show the chat capability of the diffusion language model, and the multi-turn dialogue cases are very interesting.

**Weakness**

- While one could argue that the methods used in this work including masked diffusion models and pre-training & SFT pipeline are widely studied, I find the value in training the 8B scale model with comprehensive comparisons with autoregressive models including the self-constructed AR model. Further, the showcased chat capability is a promising direction for future research on diffusion language models.

- Also one could argue that current LLaDA performance underperforms SOTA LLMs, this is not a fair comparison as the training dataset for those models is not available and the resources for training are highly incomparable.

**Questions For Authors:**

Is the inference time for LLaDA faster than the LLaMA with the same number of parameters? I would appreciate the inference time comparison and the analyses on it, for example, if faster than what component (e.g., parallel generation of tokens) makes it faster.

**Relation To Broader Scientific Literature:**

This work demonstrates the strong scalability of the diffusion language models on diverse benchmarks compared with self-constructed autoregressive language models and SOTA LLMs. This aligns with the previous findings on the scalability of the diffusion language models, for example [Nie et al., 2024].

Nie et al., Scaling up masked diffusion models on text, ICLR 2025

**Theoretical Claims:**

While there are no propositions or theorems in this work, training and inference algorithms are sound.

---

> ### Author Rebuttal · Authors · 2025-04-01
>
> # Response to Reviewer 8p3t
> We thank Reviewer 8p3t for the recognition of our contributions and the thoughtful comments. Below is our point-by-point response.
> ## Q1: Contribution
> Like most research, our work builds upon prior studies. We sincerely appreciate your recognition of our unique contributions. If you have any further questions or concerns, we would be happy to address them.
>
> ## Q2: Comparison with SOTA LLMs
> We appreciate your observation that the training data and compute used in our work are not comparable to those of SOTA LLMs. This is indeed a primary reason why LLaDA lags behind SOTA models in overall performance. We will further scale diffusion language models to better explore their full potential in future work.
>
> ## Q3: Efficiency
> We include an inference time analysis showing that **LLaDA enables a trade-off between generation quality and inference efficiency, which stems from its ability to generate multiple tokens in parallel.**
>
> We evaluate three representative benchmarks on 8 A100-80G GPUs: Math (mathematica), HumanEval (code), and MMLU (general). To highlight the efficiency potential of LLaDA, we adopt shorter generation lengths—specifically, 1 for MMLU, 128 for Math, and 256 for HumanEval. We compare LLaMA3 with and without KV-Cache, while LLaDA operates without any inference optimization techniques. Both LLaDA and LLaMA3 have 8B parameters. In the table, the numbers in parentheses indicate the number of sampling steps. **Overall, when inference time is comparable, LLaDA achieves performance similar to LLaMA3 with KV-Cache. Notably, on the Math benchmark, LLaDA even outperforms LLaMA3 with KV-Cache while using less inference time.**
>
> In addition, recent studies [1, 2, 3] have shown that distilling MDMs can greatly accelerate both text and image generation, which holds potential for improving LLaDA. We also plan to explore inference optimization techniques similar to KV-Cache to further enhance efficiency.
> ||LLaDA-Base(32)|LLaDA-Base(64)|LLaDA-Base(128)|LLaMA3 w/ Cache|LLaMA3 w/o Cache
> -|-|-|-|-|-
> Time(min)|31|61|122|79|307
> Math|12.6|18.9|22.7|15.1|15.1
>
> ||LLaDA-Base(64)|LLaDA-Base(128)|LLaDA-Base(256)|LLaMA3 w/ Cache|LLaMA3 w/o Cache
> -|-|-|-|-|-
> Time(s)|56|110|220s|342|354
> HumanEval|12.8|23.8|31.7|34.2|34.2
>
> | |LLaDA-Instruct(1)|LLaMA3 w/ Cache|LLaMA3 w/o Cache
> -|-|-|-
> Time(s)|231|235|334
> MMLU|64.5|68.4|68.4
>
> [1] Hayakawa et al. Distillation of Discrete Diffusion through Dimensional Correlations.
>
> [2] Zhu et al. DiMO: Distilling Masked Diffusion Models into One-step Generator.
>
> [3] Xu et al. Show-o Turbo: Towards Accelerated Unified Multimodal Understanding and Generation.
>
> If you have any more questions, we are happy to discuss them and will do our best to address them.

---

### Official Review · Reviewer_wFKs · 2025-03-12

**Overall Recommendation:** 3

**Summary:**

This paper focuses on the architecture of large language models (LLMs) and discusses the effectiveness of non-autoregressive training for large models. Inspired by the approach of masked language models, it designs a Masked Diffusion Language Model and proposes a diffusion model-based generative architecture with a mask predictor. The paper also explores autoregressive model architectures like LLaMA and presents a nearly identical large model training pipeline (pre-training, SFT). Through extensive experiments comparing with different AR LLMs, some pioneering conclusions are drawn. Additionally, scaling experiments demonstrate the advantages of non-autoregressive architectures in training. Finally, the paper explores some challenges encountered with the Masked Diffusion Language Model through extensive experimentation.

## update after rebuttal
Based on the review comments from all reviewers and the author's rebuttal, I believe the paper still has shortcomings. I agree with reviewer TqxC that the writing needs improvement due to an overstatement of the work. Moreover, the mask diffusion method primarily builds on existing work, lacking sufficient innovation and formula derivation. The paper primarily relies on GPU-intensive experiments for scaling, yet it does not achieve the depth expected of a technical report.

**Reasons for Acceptance**: As an experimental report on scaling, this paper indeed resolves many underlying issues and provides useful conclusions. From the perspective of community contribution, the insights generated are valuable and could justify acceptance.

**Reasons for Rejection:** As a framework for a new method, the paper falls short in efficiency and comparative experiments. The writing style does indeed exaggerate the impact, necessitating revisions.

**Claims And Evidence:**

Yes, the paper proposes that non-autoregressive architectures can train LLMs and have advantages in inference generation and scaling.

**Essential References Not Discussed:**

Currently, there is none, considering that this is the first non-autoregressive large language model and there are no non-autoregressive baselines for comparison. It is hoped that the citation section will provide a more detailed description of how existing masked diffusion language models (non-LLM) are modeled, or summarize some general modeling formulas.

**Experimental Designs Or Analyses:**

The scaling experiments seem to be conducted on models that are not sufficiently large. Observing the scaling effects on even larger models (if feasible) would provide more convincing evidence of the model's scalability. During the sampling process, the authors introduced a semi-autoregressive remasking strategy, which resulted in a noticeable decline in performance on the Instruct version of LLaDA. This aspect lacks a thorough and reasonable analysis.

**Methods And Evaluation Criteria:**

The dataset and metrics employed in this study are compliant, with a discussion on both general generative tasks and those requiring stronger reasoning capabilities. Experiments were conducted to explore the potential of masked diffusion language models. The proposed method also addresses the question that autoregressive models are not the sole approach to training large language models (LLMs), offering a new direction for training large models and uncovering their potential.

**Other Comments Or Suggestions:**

I have no comments and suggestions for the authors.

**Other Strengths And Weaknesses:**

Strengths:

1. This article is the first to conduct extensive experiments to establish a non-autoregressive large language model based on diffusion, comparing its scaling advantages with existing autoregressive LLM. It demonstrates that masked diffusion language models can scale better within a certain range, showing potential.

2. Through experiments comparing the effects of pretraining and sft with different autoregressive llms, and by using a mask predictor as a denoising network for prediction, this presents another generative approach. The experiments also highlight the model's certain advantages in some reasoning tasks.

3. This paper represents the pioneering effort in training a non-autoregressive LLMs, with its experimental design and analysis of results offering groundbreaking significance and research insights. It introduces a "new" alternative for the training of large models.

Weaknesses:

1. The experimental results, on the whole, do not significantly outperform the existing AR-LLM baselines. Although there are advantages in reasoning capabilities, it is still insufficient to claim that the proposed method is an "excellent" architecture for building LLMs. It can only be considered as one of the "effective" modeling and training options.

2. The length of the generated text is still too short and of fixed length, making it difficult to demonstrate an advantage over existing autoregressive LLMs in handling long texts. Alternatively, the potential of the proposed method could be illustrated by balancing parallel reasoning performance and generation length.

**Questions For Authors:**

1. Comparison of Inference Efficiency. Firstly, could you supplement the comparison with the inference speed of autoregressive LLMs? Since diffusion can perform parallel inference without the need for KV cache, they should theoretically be more efficient. Secondly, since the number of sampling steps in the appendix is also a hyperparameter, I would like to ask whether the proposed sampling method is compatible with algorithms designed to optimize and accelerate sampling for diffusion models, as well as reduce the number of sampling steps?

2. Performance Comparison of Non-LLMs. Could you provide the performance gap between LLaDA and previously proposed discrete or continuous diffusion LMs under the same parameter? Through comparison, it can be shown that the LLADA architecture has advantages over other non-autoregressive models, rather than previous works being able to achieve more effective scaling than LLADA.

3. Effectiveness of the Low-confidence Remask Strategy. The significant drop in LLaDA-Instruct performance in Table 6 seems somewhat unreasonable. I speculate that this strategy might disrupt the continuity of the sampling process (in other words, diffusion path), such as increasing the difficulty of denoising for the mask predictor. Directly applying this strategy in SFT model may not necessarily be effective, and I hope that some theoretical evidence can be provided to demonstrate that the Low-confidence Remasking Strategy can be effectively applied to LLaDA. Moreover, since the remask strategy is used every time during text generation sampling, I wonder what impact this has on the generation speed.

4. Discussion on Semi-Autoregressive and Block Experiments. Semi-autoregressive methods are only used in the remask strategy; could blocks be input to the mask predictor for generation? Additionally, due to the limitations on generation length, the size and number of blocks also require more experiments to substantiate.

**Relation To Broader Scientific Literature:**

The authors have summarized the applications of discrete and continuous diffusion models in text generation, and the proposed scaling and training of an LLM from scratch are deemed reasonable.

**Theoretical Claims:**

This paper is an experimental study, and the proposed masked diffusion language model is theoretically supported, as well as being relatively simple and comprehensive.

---

> ### Author Rebuttal · Authors · 2025-04-01
>
> # Responses to reviewer wFKs
> We thank Reviewer wFKs for the recognition of our contributions and the thoughtful comments. Below is our point-by-point response.
> ## Q1: AR baselines
> We agree that the current results may still be insufficient to claim an "excellent" architecture for building LLMs, and we appreciate your acknowledgment that our method is one of the "effective" modeling and training options.
>
> We believe that challenging the long-held assumption that LLMs must rely on autoregressive training is a meaningful contribution in itself. Our work is an early attempt to scale diffusion language models, with many design choices—such as data and architecture—borrowed from autoregressive settings. In future work, we aim to develop designs specifically tailored to diffusion models to further enhance their performance.
> ## Q2: Long texts
> Since LLaDA is built upon the Transformer architecture, many existing techniques [1, 2] for long-context processing in Transformers are applicable to LLaDA, and [1] has already been integrated. We leave a more thorough exploration of LLaDA’s long-context capabilities to future work.
>
> [1] Su et al. RoFormer: Enhanced Transformer with Rotary Position Embedding.
>
> [2] Press et al. Train Short, Test Long: Attention with Linear Biases Enables Input Length Extrapolation.
> ## Q3: Efficiency
> Please refer to our response to Q1 from Reviewer TqxC for efficiency analysis. **We show that LLaDA enables a trade-off between generation quality and inference speed, and achieves performance comparable to LLaMA3 with KV-Cache when inference time is similar. Notably, on the Math benchmark, LLaDA even outperforms LLaMA3 with KV-Cache while using less inference time.**
>
> Recent studies [3, 4] have demonstrated that distillation can significantly accelerate both text and image generation with discrete diffusion models. These approaches hold strong potential for improving LLaDA, and we leave their integration to future work.
>
> [3] Hayakawa et al. Distillation of Discrete Diffusion through Dimensional Correlations.
>
> [4] Zhu et al. DiMO: Distilling Masked Diffusion Models into One-step Generator.
> ## Q4: Comparison of Non-LLMs
> LLaDA is built on RADD [5], one of the best-performing discrete diffusion models [6, 7], which has been shown to outperform continuous diffusion models at comparable parameter scales (~0.3B). Due to limited computational resources, we did not extend this comparison to larger model sizes.
>
> However, these previous works use small models, lack key LLM capabilities such as in-context learning and instruction-following, and are evaluated only on perplexity rather than downstream tasks.
>
> In contrast, our work is the first to scale diffusion language models to unprecedented 8B parameters, demonstrating better scalability than autoregressive models while supporting these core LLM abilities. As you noted, *“this is the first non-autoregressive large language model and there are no non-autoregressive baselines for comparison.”*
>
> [5] Ou et al. Your Absorbing Discrete Diffusion Secretly Models the Conditional Distributions of Clean Data.
>
> [6] Sahoo et al. Simple and Effective Masked Diffusion Language Models.
>
> [7] Shi et al. Simplified and Generalized Masked Diffusion for Discrete Data.
> ## Q5: Lowest-confidence remasking
> Applying lowest-confidence remasking directly to LLaDA-Instruct may lead the model to generate excessive |EOS| tokens, resulting in overly short answers. This is the primary cause of the observed performance degradation.
>
> Lowest-confidence remasking is a heuristic strategy similar to the widely used annealed sampling in LLMs, which emphasizes high-probability tokens. During SFT, we pad |EOS| tokens within each mini-batch to align sequence lengths, increasing their frequency in the training data. As a result, the remasking strategy tends to select |EOS| more often. We plan to address this in future work by improving the SFT data processing pipeline.
>
> Regarding generation speed, the remasking strategy has negligible impact on runtime efficiency. It introduces no additional neural network computation and only involves lightweight indexing operations with minimal overhead.
> ## Q6: Semi-AR
> Thank you for your suggestion regarding feeding blocks into the mask predictor. This would require modifying the training procedure of LLaDA, and while we believe it could be beneficial for inference, we leave it as future work.
>
> As shown in our response to Q5 from Reviewer OmMG, we conducted experiments with block lengths of 4, 8, 16, 32, and 64. The results are consistently robust across these configurations.
> ## Q7: Scaling to larger models
> We agree that scaling to larger models is a valuable direction and we leave large-scale experiments for future work.
> ## Q8: Summarizing previous MDMs
> We will include a summary of the general modeling formulations of previous MDMs in the revision.
>
> If you have any more questions, we are happy to discuss them and will do our best to address them.

---

> > ### Comment · Reviewer_wFKs · 2025-04-02
> >
> > Thank you for your responses. I have carefully read your responses, and they addresse the vast majority of the concerns. Given that my score is already the highest, and that it seems appropriate, I'm keeping it for now.

---

> > > ### Author Response · Authors · 2025-04-05
> > >
> > > We thank Reviewer wFKs for acknowledging our contributions since the beginning. We are glad that the vast majority of the concerns have been addressed.

---

### Official Review · Reviewer_QmMG · 2025-03-12

**Overall Recommendation:** 3

**Summary:**

The paper introduces LLaDA, a large language model based on diffusion models instead of autoregressive models (ARMs), which dominate in the large language modeling area currently. The model is trained from scratch using a pre-training and supervised fine-tuning (SFT) paradigm with a mask diffusion loss, achieving competitive performance with strong LLMs like LLaMA3 8B. The authors show LLaDA's scalability, in-context learning, and instruction-following on various tasks.

**Claims And Evidence:**

Yes.

**Essential References Not Discussed:**

No.

**Experimental Designs Or Analyses:**

Yes. The benchmarks used to evaluate an LLM and the experimental analyses are reasonable.

**Methods And Evaluation Criteria:**

Yes.

**Other Comments Or Suggestions:**

No.

**Other Strengths And Weaknesses:**

Strengths:
- The paper introduces a distinct approach to large language modeling using diffusion models, which is a significant departure from the dominant autoregressive paradigm.
- The trained LLaDA with 8B parameter size demonstrates competitive performance with state-of-the-art LLMs like LLaMA3 8B, and excels in math, infilling, and reversal reasoning tasks particularly.

Weaknesses:
- The method part seems to mostly follow [1] and [2], which makes this work appear to merely scale the model parameters without providing any new methods or insights. Though scaling alone is already a valuable contribution in some way.
- The text diffusion model seems to require significant computational resources during inference, which may limit its practical applicability compared to more efficient ARMs with KV cache.

[1] Shen Nie et al. Scaling up masked diffusion models on text. arXiv preprint arXiv:2410.18514, 2024.

[2] Subham Sekhar Sahoo et al. Simple and effective masked diffusion language models. arXiv preprint arXiv:2406.07524, 2024.

**Questions For Authors:**

- I'm curious about how a diffusion model does code infilling as the middle length seems to need to be decided in advance per my understanding. Could the authors elaborate more on how HumanEval-FIM is evaluated in LLaDA?
- In Table 6, why the semi-autoregressive remasking is significantly effective on the instruct model but less effective for the base model?
- In Table 8, how do we decide the block length for each task in advance?

**Relation To Broader Scientific Literature:**

Autoregressive models (ARMs) are widely regarded as the cornerstone of large language models (LLMs), while this paper tries to challenge this by using a diffusion modeling objective to train LLMs.

**Theoretical Claims:**

Yes. I've checked all the equations in the paper and didn't notice any significant errors.

---

> ### Author Rebuttal · Authors · 2025-04-01
>
> # Responses to Reviewer OmMG
> We thank Reviewer OmMG for the recognition of our contributions and the thoughtful comments. Below is our point-by-point response.
>
> ## Q1: Contributions
> Thank you for recognizing our efforts in scaling masked diffusion models to an unprecedented 8B scale. Like most prior work, our research builds on established foundations. We believe a key open question for masked diffusion models is whether they can scale comparably to autoregressive models while exhibiting core capabilities such as in-context learning and instruction following—abilities previously considered unique to autoregressive approaches. We view this line of investigation as equally important as algorithmic innovation.
>
>
> ## Q2: Efficiency
> Please refer to our response to Q1 from Reviewer TqxC, where we provide a sampling efficiency comparison with LLaMA3. **We show that LLaDA enables a trade-off between generation quality and inference speed, and achieves performance comparable to LLaMA3 with KV-Cache when inference time is similar. Notably, on the Math benchmark, LLaDA even outperforms LLaMA3 with KV-Cache while using less inference time.**
>
> While our work focuses on exploring the upper bound of masked diffusion models for language modeling, we believe there is still significant room to improve sampling efficiency through further optimization.
>
> Recent studies [1, 2, 3] have demonstrated that distillation can significantly accelerate both text and image generation with masked diffusion models. In addition, [4] shows that masked diffusion models can leverage KV-Cache. These techniques hold potential for improving LLaDA.
>
> [1] Hayakawa et al. Distillation of Discrete Diffusion through Dimensional Correlations.
>
> [2] Xu et al. Show-o Turbo: Towards Accelerated Unified Multimodal Understanding and Generation.
>
> [3] Zhu et al. DiMO: Distilling Masked Diffusion Models into One-step Generator.
>
> [4] Arriola et al. Block Diffusion: Interpolating Between Autoregressive and Diffusion Language Models.
>
>
> ## Q3: Evaluate on HumanEval-FIM
> Similar to autoregressive models, for HumanEval-FIM, we provide the prefix and suffix as the prompt and require the model to complete the missing middle part at the end of the sequence. We will include a detailed explanation in the revision.
>
>
> ## Q4: Semi-AR
> As detailes in Appendix B.3, for LLaDA-Instruct, semi-autoregressive remasking helps prevent overly short outputs caused by padded |EOS| tokens in the SFT data. In contrast, the pre-training data used for LLaDA-Base does not include padded |EOS| tokens, so semi-autoregressive remasking is unnecessary.
>
> Specifically, during SFT, we pad |EOS| tokens within each mini-batch to align sequence lengths. When using the lowest-confidence remasking strategy, which is similar to annealed sampling in LLMs and places greater weight on high-probability tokens, LLaDA-Instruct tends to overgenerate the |EOS| token due to its frequent occurrence in the SFT data. This often results in outputs that are too short.
>
> Semi-autoregressive remasking mitigates this issue by prioritizing the prediction of content tokens in the earlier part of the answer, rather than allowing the model to prematurely focus on predicting the |EOS| token. This helps produce longer, more complete responses. In future work, we plan to address this issue by optimizing the SFT process.
>
> ## Q5: Block length
> We can choose the block length from {4, 8, 16, 32, 64}, and the results remain highly robust across these settings. The corresponding GSM8K results for different block lengths are as follows:
> | | 4| 8 | 16 | 32 | 64|
> |-|-|-|-|-|-|
> |GSM8K results|78.5| 78.6| 78.2 | 77.5| 77.8|
>
> If you have any more questions, we are happy to discuss them and will do our best to address them.

---

### Official Review · Reviewer_TqxC · 2025-03-14

**Overall Recommendation:** 1

**Summary:**

This paper scales up discrete diffusion models to a larger regime than has been seen in prior work (8B params, 2.3T tokens) and additionally perform supervised instruction tuning. They present discrete diffusion models as an alternative to autoregressive language modeling. They evaluate their discrete diffusion model across natural language understanding, coding, and mathematics benchmarks and show strong results compared to autoregressive models at a similar scale. They also present unique benefits of discrete diffusion modeling, such as breaking the "reversal curse" observed with causal language models. For generation, they explore different decoding algorithms such as semi-autoregressive unmasking and lowest confidence unmasking.

**Claims And Evidence:**

I do not think the central claim that diffusion models are "a viable and promising alternative to ARMs" is well-supported by the presented evidence, primarily due to the lack of any inference-time analysis.

The paper shows that LLaDA achieves competitive performance on certain tasks, but the authors completely omit any analysis of generation time or inference efficiency. Diffusion models can naturally trade-off compute time for quality (the authors use between 64-1024 steps as shown in Figure 5). Understanding this trade-off, and where autoregressive models of a comparable scale fall on that tradeoff curve, is critical for evaluating ther primary claim.

The authors do demonstrate that their pure diffusion sampling (random remasking) achieves reasonable performance on some tasks. The pre-trained model achieves 52.3% accuracy on GSM8K with this approach. However, their generation results for their final model typically come from their "Lowest confidence & semi-autoregressive remasking" strategy, which incorporates autoregressive elements by proceeding left-to-right in blocks.

It's not clear whether diffusion models can match autoregressive models on these tasks without borrowing autoregressive elements. Given the framing in the introduction challenging autoregressive generation, I would expect pure diffusion generation to be evaluated comprehensively. They do present a full ablation on GSM8k, but not the other generation datasets.

For the NLU benchmarks, the authors apply classifier-free guidance for LLaDA but not autoregressive baselines. The same technique has been shown to be beneficial to autoregressive models as well [1]. Result should be reported without such techniques for both methods. If the guidance strength is swept over for the diffusion model, then the same should be done for the autoregressive model.

Because of these issues, the evidence falls short of supporting the claim that diffusion models represent a generally viable alternative to autoregressive approaches.

[1] Sanchez, Guillaume, et al. "Stay on Topic with Classifier-Free Guidance." International Conference on Machine Learning. PMLR, 2024.

**Essential References Not Discussed:**

The discussion of related work is reasonable.

**Experimental Designs Or Analyses:**

The experimental design has several methodological issues that affect the validity of the comparisons:

1) The authors apply classifier-free guidance to improve LLaDA's performance but don't apply the technique to autoregressive baselines, creating an uneven comparison. Results without such techniques should be reported.

2) While the paper ablates their remasking strategies for GSM8K (Table 6), the results across generation tasks tasks rely on their "Lowest confidence & semi-autoregressive remasking" strategy, which incorporates autoregressive elements. A more comprehensive comparison of pure diffusion approaches across all tasks would have strengthened the experimental design, especially given their claim that autoregression is not necessary for the capabilities of currnent LLMs.

3) For a study proposing an alternative to autoregressive models, the absence of generation time comparisons makes evaluating the viability of their model challenging.

**Methods And Evaluation Criteria:**

The proposed method of scaling up masked diffusion models is reasonable for exploring alternatives to autoregressive language modeling. The authors build on existing masked diffusion techniques and successfully scale them to 8B parameters.

The evaluation criteria generally make sense, as the authors use standard benchmarks (MMLU, GSM8K, HumanEval, etc.) that allow for comparison with existing autoregressive LLMs. Their inclusion of reversal reasoning tasks is appropriate given the potential advantages of bidirectional modeling compared to autoregressive modeling.

However, there is no inference-time efficiency analysis in this work. For a paper proposing an alternative to autoregressive models, understanding the computational requirements during generation is critical. Without quantifying this tradeoff, the evaluation does not clearly present the tradeoffs between approaches.

Additionally, the ablation studies would have been stronger if they had compared pure diffusion approaches consistently across all tasks, rather than primarily for GSM8K.

While I recognize the value of comparing on various downstream tasks instead of focusing the evaluation on validation likelihood/perplexity, I think the inclusion of those results for the autoregressive and discrete diffusion models would be informative.

**Other Comments Or Suggestions:**

No additional comments.

**Other Strengths And Weaknesses:**

Strengths:

1. Successfully demonstrates that diffusion models can scale to 8B parameters and achieve competitive performance on standard language tasks

2. Shows potential advantages in bidirectional reasoning, particularly in reversal tasks

3. Provides a comprehensive empirical evaluation across multiple benchmark tasks

Weaknesses:

1. The paper uses a grandiose writing style that detracts from the scientific content. Literary quotes from William Blake ("What is now proved was once only imagined") and Albert Einstein ("In the middle of difficulty lies opportunity") are inappropriate for a scientific paper. Similarly, the use of ill-defined language such as "intelligence" harms clarity.

2. The paper places strange emphasis on their use of maximum likelihood training as though it represents a novel contribution. The authors simply train their model to maximize the likelihood of the data (or a lower bound on it). This is the approach used by most generative models, including recent discrete diffusion models and autoregressive LLMs. The discussion of "Fisher consistency" and connection to data compression is spurious.

3. Although this paper is primarily an engineering effort, the paper provides limited documentation of things such as the pre-training data composition, SFT data curation, etc. that would enable others to reproduce or build upon this work.

**Questions For Authors:**

1. What is the inference-time of your method compared to the baseline autoregressive model? For instance, across the sampling step sweep you present on GSM8k in Figure 5.
2. What is the performance of your approach on the NLU benchmarks without the use of classifier-free guidance?

**Relation To Broader Scientific Literature:**

This paper primarily scales up existing masked diffusion models (MDMs) to 8B parameters. The approach builds on prior work in discrete diffusion model (eg. Austin et al. (2021), Ou et al. (2024)).

The authors apply the established pre-training and supervised fine-tuning (SFT) paradigm that has become standard for autoregressive LLMs to discrete diffusion models.

The paper's main contribution is demonstrating that MDMs can scale effectively and achieve competitive performance with similar-sized autoregressive models on standard benchmarks. This extends earlier work by Nie et al. (2024), which explored the scaling behavior of masked diffusion at a much smaller scale.

They evaluate whether discrete diffusion perform better on reversal tasks which is related to prior work on the "reversal curse" in autoregressive models (Berglund et al. (2023)). They provide empirical evidence that bidirectional modeling may address this limitation.

**Theoretical Claims:**

The paper does not present any novel proofs.

---

> ### Author Rebuttal · Authors · 2025-04-01
>
> # Responses to Reviewer TqxC
> We thank Reviewer TqxC for the thoughtful comments. Below is our point-by-point response.
> ## Q1: Efficiency
> We include an inference time analysis showing that LLaDA enables a trade-off between generation quality and inference efficiency.
>
> We evaluate three representative benchmarks on 8 A100-80G GPUs: Math (mathematica), HumanEval (code), and MMLU (general). To highlight the efficiency potential of LLaDA, we adopt shorter generation lengths—specifically, 1 for MMLU, 128 for Math, and 256 for HumanEval. We compare LLaMA3 with and without KV-Cache, while LLaDA operates without any inference optimization techniques. In the table, the numbers in parentheses indicate the number of sampling steps. **Overall, when inference time is comparable, LLaDA achieves performance similar to LLaMA3 with KV-Cache. Notably, on the Math benchmark, LLaDA even outperforms LLaMA3 with KV-Cache while using less inference time.**
>
> Recent studies [1] have shown that distilling MDMs can greatly accelerate both text and image generation, which holds potential for improving LLaDA. We also plan to explore inference optimization techniques similar to KV-Cache to further enhance efficiency.
> ||LLaDA-Base(32)|LLaDA-Base(64)|LLaDA-Base(128)|LLaMA3 w/ Cache|LLaMA3 w/o Cache
> -|-|-|-|-|-
> Time(min)|31|61|122|79|307
> Math|12.6|18.9|22.7|15.1|15.1
>
> ||LLaDA-Base(64)|LLaDA-Base(128)|LLaDA-Base(256)|LLaMA3 w/ Cache|LLaMA3 w/o Cache
> -|-|-|-|-|-
> Time(s)|56|110|220s|342|354
> HumanEval|12.8|23.8|31.7|34.2|34.2
>
> | |LLaDA-Instruct(1)|LLaMA3 w/ Cache|LLaMA3 w/o Cache
> -|-|-|-
> Time(s)|231|235|334
> MMLU|64.5|68.4|68.4
>
> [1] Hayakawa et al. Distillation of Discrete Diffusion through Dimensional Correlations.
> ## Q2: Semi-AR
> We clarify the misunderstanding on *"their generation results for their final model typically come from their 'Lowest confidence & semi-autoregressive remasking' strategy"*. As detailed in Lines 1072-1087, **20 out of 24 tasks in Tables 1&2 used pure diffusion strategy without Semi-AR for likelihood evaluation and sampling.** Further, we add pure diffusion sampling results for the remaining four tasks:
> ||GSM8K|GPQA|HumanEval|MBPP
> -|-|-|-|-
> w/o Semi-ar|62.9|30.3|43.9|28.0
> w/ Semi-ar|78.6|31.8|47.6|34.2
>
> **Our main conclusions remain unchanged without Semi-AR.** The evaluation of LLaDA-Base involves no Semi-AR and achieves performance comparable to LLaMA3-Base. And the conclusion that LLaDA-Instruct slightly underperforms LLaMA3-Instruct (Line 266) also holds. This relative underperformance is attributed to the absence of alignment with RL, which we leave for future work.
>
> We also compared pure diffusion and Semi-AR evaluation across all 24 tasks, and found that Semi-AR yielded improvements on only above 4 tasks. Due to space constraints, both sets of results will be included in Tables 1&2 in the final version.
> ## Q3: CFG
> As detailed in Lines 1081–1087, 18 out of the 24 benchmarks did not use CFG. We add the results without CFG for the remaining 6 benchmarks:
> ||ARC-C|Hellaswag|TruthfulQA|WinoGrande|GPQA|PIQA
> -|-|-|-|-|-|-
> LLaDA-Base w/ CFG|47.9|72.5|46.4|74.8|26.1|74.4
> LLaDA-Base w/o CFG|45.9|70.5|46.1|74.8|25.2|73.6
>
> After removing CFG, all performance drops are within 2 points. These changes do not affect our conclusion that LLaDA-Base is comparable to LLaMA3-Base. We will include both with and without CFG results in the revised version and provide a discussion of the reference you mentioned.
> ## Q4: PPL
> We add the zero-shot perplexity results:
> ||WikiText2|WikiText103|LAMBADA
> -|-|-|-
> LLaDA-Base|11.4|11.3|26.3
> LLaMA3-Base|8.1|8.2|19.9
> LLaMA2-Base|37.3|36.6|63.2
> ## Q5: Writing style
> We appreciate your suggestions and we will carefully revise the paper to improve clarity.
> ## Q6: MLE
> We clarify that **maximum likelihood estimation (MLE) is not our contribution but rather our motivation.** As stated in the introduction, we consider MLE a key factor in the success of LLMs, which motivates us to scale MDMs and develop LLaDA. Further, we will revise the discussion on data compression and Fisher consistency, and include their theoretical foundations below.
>
> As detailed in the background of [1], MLE is equivalent to lossless data compression under Shannon's source coding theorem.
>
> As detailed on page 39 of [2], Fisher consistency refers to an estimation method that recovers the true parameter when applied to the entire population and MLE satisfies Fisher consistency.
>
> [1] Delétang et al. Language Modeling Is Compression.
>
> [2] R. H. Norde. A Survey of Maximum Likelihood Estimation.
> ## Q7: Data
> Our pre-training data is sourced from the public Common Crawl, comprising approximately 11% Chinese, 61% English, and 28% code. Our SFT dataset includes 1M open-source samples and 3.5M synthetic samples. We will provide details of our data filtering and synthesis pipeline in the revision, with a level of detail comparable to that of LLaMA3.
>
> If you have any more questions, we are happy to discuss them and will do our best to address them.

---

> > ### Comment · Reviewer_TqxC · 2025-04-07
> >
> > Thank you for your response and the additional results.
> >
> > Q1. Thank you for providing the timing comparisons. Given the centrality of inference efficiency to the motivation of discrete diffusion models, a comprehensive comparison, beyond what can fit in a rebuttal, would be needed to support the strong claims put forth in the paper about the competitiveness of discrete diffusion generation with autoregressive generation. When using the same number of "steps" as an autoregressive model, discrete diffusion models are much slower due to the lack of KV-caching (e.g. see [1, 2]). Therefore it is difficult to understand how LLaDA-Base(256) is faster than LLaMA3 w/ Cache given that LLaDA-Base(256) is using one timestep per token in the generation sequence. I understand that full details about the timing comparison likely did not fit in the rebuttal response. However, this is why presenting a detailed analysis in the original submission is critical.
> >
> > The model as currently presented (i.e. employing bidirectional attention at every timestep) is inherently incompatible with KV cacheing. The use of such techniques could be enabled by explictly training the model in a block-wise autoregressive fasion (as was done in past work [3]), but that is a significant departure from the current work.
> >
> > [1] Sahoo et al. "Simple and effective masked diffusion language models." NeurIPS (2024).
> >
> > [2] Zheng et al. "Masked diffusion models are secretly time-agnostic masked models and exploit inaccurate categorical sampling." arXiv preprint (2024).
> >
> > [3] Arriola et al. "Block diffusion: Interpolating between autoregressive and diffusion language models." arXiv preprint (2025).
> >
> > Q2. When looking at the sampling configurations for the final LLaDA-Instruct model (Table 8), 3/4 open-ended generation tasks (meaning not multiple choice QA) employ semi-autoregressive generation with a block size significantly smaller than the overall sequence length. I am referring specifically to GSM8K, HumanEval, and MBPP. I acknowledge that the MATH benchmark is open-ended generation as well and does not employ semi-autoregressive generation.
> >
> > The NLU benchmarks only require computing likelihoods and therefore do not involve actual generation. As a result, I think that my original statement was accurate. I recognize that numerous benchmarks that are not open-ended generation are also reported, but I think placing emphasis on the generation behavior of the language model is reasonable given that is how they are used in practice.
> >
> > While I appreciate these additional results, they actually reinforce my original concern rather than addressing it. Your updated results do show meaningful performance drops from disabling semi-autoregressive generation (GSM8K: 78.6 -> 62.9, HumanEval: 47.6 -> 43.9, MBPP: 34.2 -> 28.0) which conflicts with the presented claim in the paper that the autoregressive formulation is not essential for the success of language models.
> >
> > Q3. Thank you for providing the additional results. I acknowledge that the model performance is still reasonably strong without the use of CFG, although it did provide some benefit.
> >
> > Q4. Perplexity is not directly comparable across models with different vocabularies. As a result, it's not clear that these results are meaningful.
> >
> > Q6: As written, it is presented as something unique to this work. For instance, the introduction states (emphasis mine):
> >
> > > This design enables LLaDA to construct a model distribution with bidirectional dependencies and optimize a lower bound of its log-likelihood, offering an **unexplored** and principled alternative to existing LLMs.
> >
> > Again in the conclusion:
> >
> > > We introduce LLaDA, a principled and **previously unexplored approach** to large language modeling based on diffusion models.
> >
> > This is simply untrue. There is a significant body of work exploring exactly this class of generative models for language generation. The contribution of this work is in scaling up an existing approach, not introducing a fundamentally new paradigm.
> >
> > Q7. I appreciate the additional details about the data curation process. For the engineering effort to be a meaningful contribution to the research community in itself, the level of detail and commitment to open-source by the OLMo [4] line of models would be a better benchmark. The level of detail presented by the Llama3 technical report is still relatively limited.
> >
> > [4] OLMo: Accelerating the Science of Language Models (Groeneveld et al., ACL 2024)
> >
> > In summary, I think that this is an impressive scaling effort of an interesting approach to language generation. However, the claims in the paper, as written, are very strong and the reported results do not support those claims. A rigorous analysis of inference-time efficiency is necessary and the claims about the unessential nature of autoregression should be softened as long as the strongest results require semi-autoregressive generation.

---

> > > ### Author Response · Authors · 2025-04-09
> > >
> > > Thank you for your response. Although we **received it less than two days before the rebuttal deadline**, and the guidelines allow authors to skip the final response if insufficient time is given, we still did our best to address your comments as follows:
> > >
> > > Q1. As detailed in Lines 103–105 (left) and 55–67 (right), we claim that LLaDA is comparable to large-scale (8B) ARMs in scalability and other performance metrics, with sufficient validation. We **disagree, with our highest respect**, with the implication that comparable performance implies comparable efficiency. We make no claims suggesting LLaDA outperforms ARMs in efficiency.
> > >
> > > For completeness, our **original submission** already included a trade-off analysis between NFE and performance (Figure 5). In this rebuttal, we added an additional analysis on sampling time vs. performance **in response to your suggestion**. The works you cited also conducted similar comparisons with ARMs—specifically, Figure 2 in [1], Figure 9 in [2], and Table 7 in [3].
> > >
> > > **We clarify that LLaMA3’s output length is not controlled due to its autoregressive nature.** On HumanEval and MMLU, its average output lengths are 433 and 2, compared to LLaDA’s 256 and 1, contributing to LLaDA’s faster performance. The references [1, 2, 3] you mentioned evaluate generation quality only using perplexity. We go a step further by using three downstream tasks as evaluation metrics.
> > >
> > > In summary, **we believe that the inference efficiency of LLaDA has been thoroughly analyzed**, even though this does not affect our main conclusions. Besides, **we kindly ask that you not assume we are unable to address your concerns within the rebuttal period**.
> > >
> > > Q2. We emphasize that **we follow existing LLMs (e.g. LLaMA series) to comprehensively evaluate LLaDA-Base/Instruct on common benchmarks** and our claim remains **the overall performance (not just open-ended generation) of LLaDA is competitive to ARMs**.
> > >
> > > **As you stated in the rebuttal comment, your concerns refer specifically to open-ended generation tasks,** we conducted a more in-depth investigation into the sampling strategy for such tasks. We set the probability of the |EOS| token to zero during the LLaDA-Instruct's sampling. Note that in LLaDA-Instruct, |EOS| is used for padding purposes, while the actual end-of-sequence is indicated by a different token |EOS-ids|. Please refer to our response to Q4 from Reviewer OmMG for the motivation. **This adjustment shows the potential of pure diffusion sampling, outperforming semi-AR on three of four open-ended generation tasks.**
> > > ||GSM8K|HumanEval|MBPP|Math
> > > -|-|-|-|-
> > > w/o Semi-ar|66.0|49.4|37.6|26.6
> > > w/ Semi-ar|78.6|47.6|34.2|23.8
> > >
> > > For the remaining 20 tasks, pure diffusion is competitive with semi-AR as follows:
> > >
> > > LLaDA-Base
> > > ||MMLU|BBH|ARC–C|Hella-swag|TruthfulQA|WinoGrande|PIQA|GSM8K|Math|GPQA|HumanEval|HumanEval-FIM|MBPP|CMMLU|C–Eval
> > > -|-|-|-|-|-|-|-|-|-|-|-|-|-|-|-
> > > w/o Semi-AR|**65.9**|**49.8**|45.9|**70.5**|**46.1**|**74.8**|73.6|**70.7**|**27.3**|**25.2**|**33.5**| **73.8**|38.2|**69.9**|**70.5**|
> > > w/ Semi-AR|**65.9**|48.6|**46.2**|45.2|45.2|74.3|**76.3**|70.6|27.2|24.1|33.1|73.7|**39.0**|**69.9**| **70.5**|
> > >
> > > LLaDA-Instruct
> > > ||MMLU|MMLU-pro|Hellaswag|ARC-C|GPQA
> > > -|-|-|-|-|-
> > > w/o Semi-AR|**65.5**|**37.0**|**74.6**|**88.5**|30.3
> > > w/ Semi-AR|65.4|34.8|73.3|85.4|**31.8**
> > >
> > >  **We emphasize that these experiments are conducted only following your suggesions and do not change the claim of our submission.**
> > >
> > > Q3. We're glad the CFG issue is resolved.
> > >
> > > Q4. We added perplexity experiments for LLaDA and LLaMA, as your initial comment didn’t specify the experimental setup. Line 155 explains the vocabulary difference from LLaMA. If needed, we’re happy to compare with our ARM baseline under the same vocabulary in the final version.
> > >
> > > Q6. We disagree with our highest respect. Our claim refers to the full statement: **a principled and previously unexplored approach to large language modeling**. Previous works have not explored approaches to **large language models.** We believe this aligns with your comment that *this is an impressive scaling effort of an interesting approach to language generation*. If you have any constructive suggestions on how to improve the wording, we will carefully consider them.
> > >
> > > Q7. We will open source the data collection process, model weights, evaluation code, and core training code. **We believe our level of openness is no less than that of most leading LLM papers**, including representative works such as the LLaMA series.
> > >
> > > We are confident that we have thoroughly addressed your concerns regarding semi-AR sampling and inference efficiency, as we did with your comments on CFG. In light of your assessment of our work as **impressive and interesting**, we earnestly ask you to reconsider whether a score of 1 is still warranted.
> > >
> > > **Provided that it does not violate the conference policy, we intend to publicly release the full review history after the final decision.**

---

### Decision · Program_Chairs · 2025-05-01

**Decision:**

Reject

**Comment:**

This submission presents LLaDA, a large-scale masked diffusion language model (8B parameters), claiming it as a viable alternative to autoregressive models like LLaMA3, with competitive performance on benchmarks like MMLU, GSM8K, and HumanEval, and advantages in reversal reasoning. Strengths include its pioneering effort in scaling diffusion models to this size, demonstrating core LLM capabilities (in-context learning, instruction-following), and providing a novel perspective on non-autoregressive architectures. However, weaknesses lie in overstated claims of paradigm-shifting novelty, insufficient inference-time efficiency analysis critical for diffusion models, and reliance on semi-autoregressive strategies for key results, which undermines the non-autoregressive claim. The submission lacks rigorous efficiency comparisons and deeper methodological innovation beyond scaling, limiting its impact.

In the rebuttal, authors addressed concerns by adding efficiency analyses and pure diffusion results, showing LLaDA’s trade-offs and improved non-autoregressive performance on three of four open-ended tasks. Reviewers still raised persistent issues in either reply or AC-reviewer discussion: one mentioned the need for a more comprehensive efficiency analysis given the model’s framing, and questioned novelty claims as scaling existing work. Another reviewer found efficiency comparisons inadequate and novelty overstated, while others valued the scaling contribution but agreed on exaggerated framing. Post-rebuttal, one reviewer dropped from accept (4) to weak accept (3), citing overstated claims, while others maintained borderline or accept scores. The semi-AR issue is addressed by the authors in the additional results, but the reviewers note that the original submission heavily relies on it and more inference-time analysis would still be needed. This is a borderline paper. I read the message from the authors and carefully read through all the rebuttal discussions, and I generally agree with the reviewers. In my final recommendation, I weighed the lack of original efficiency analysis and consensus of paradigm-challenging overclaims heavily, as they weaken the core thesis, against the valuable scaling insights. I give weak reject recommendation. With outstanding issues addressed this work can be impactful.